# Beyond Deep Ensembles: A Large-Scale Evaluation of Bayesian Deep Learning under Distribution Shift

**Florian Seligmann**\*
Karlsruhe Institute of Technology

**Philipp Becker**
Karlsruhe Institute of Technology
FZI Research Center for Information Technology

**Michael Volpp**
Karlsruhe Institute of Technology
Bosch Center for Artificial Intelligence

**Gerhard Neumann**
Karlsruhe Institute of Technology
FZI Research Center for Information Technology

## Abstract

Bayesian deep learning (BDL) is a promising approach to achieve well-calibrated predictions on distribution-shifted data. Nevertheless, there exists no large-scale survey that evaluates recent SOTA methods on diverse, realistic, and challenging benchmark tasks in a systematic manner. To provide a clear picture of the current state of BDL research, we evaluate modern BDL algorithms on real-world datasets from the WILDS collection containing challenging classification and regression tasks, with a focus on generalization capability and calibration under distribution shift. We compare the algorithms on a wide range of large, convolutional and transformer-based neural network architectures. In particular, we investigate a signed version of the expected calibration error that reveals whether the methods are over- or underconfident, providing further insight into the behavior of the methods. Further, we provide the first systematic evaluation of BDL for fine-tuning large pre-trained models, where training from scratch is prohibitively expensive. Finally, given the recent success of Deep Ensembles, we extend popular single-mode posterior approximations to multiple modes by the use of ensembles. While we find that ensembling single-mode approximations generally improves the generalization capability and calibration of the models by a significant margin, we also identify a failure mode of ensembles when finetuning large transformer-based language models. In this setting, variational inference based approaches such as last-layer Bayes By Backprop outperform other methods in terms of accuracy by a large margin, while modern approximate inference algorithms such as SWAG achieve the best calibration.

## 1 Introduction

Real-world applications of deep learning require accurate estimates of the model's predictive uncertainty [2, 21, 38]. This is particularly relevant in safety-critical applications of deep learning, such as medical applications [95] and self-driving cars [27]. Therefore, we want our models to be *calibrated*: A model should be confident about its prediction if and only if the prediction will likely be correct. Only then it is sensible to rely on high-confidence predictions, and, e.g., to contact a human expert in the low-confidence regime [4]. Calibration is particularly relevant when models are evaluated on out-of-distribution (o.o.d.) data, i.e. on inputs that are very different from the training data, and hence, the model cannot always make accurate predictions. However, typical deep neural networks are highly overconfident on o.o.d. data [28, 72].

---

\*Correspondence to florian.seligmann@student.kit.edu.

37th Conference on Neural Information Processing Systems (NeurIPS 2023).

Bayesian deep learning (BDL) promises to fix this overconfidence problem by marginalizing over the posterior of the model's parameters. This process takes all explanations that are compatible with the training data into account. As desired, explanations will disagree on o.o.d. data, so that predictions will have low confidence in this regime. While computing the exact parameter posterior in BDL is infeasible, many approximate inference procedures exist to tackle this problem, aiming at making BDL applicable to real-world problems. Yet, recent BDL algorithms are typically only evaluated on the comparatively small and curated MNIST [52], UCI [16], and CIFAR [48] datasets with artificial o.o.d. splits. Existing BDL surveys [19, 20, 29, 72] concentrate on a few popular but relatively old algorithms such as Bayes By Backprop, Deep Ensembles, and Monte Carlo Dropout. In the light of recent calls for more realistic benchmarks of state-of-the-art (SOTA) algorithms [1] – with some experts going as far as calling the current state of BDL a "replication crisis"[2] – we aim to provide a large-scale evaluation of recent BDL algorithms on complex tasks with large, diverse neural networks.

**Contributions.** **i)** We systematically evaluate a comprehensive selection of modern, scalable BDL algorithms on large image- and text-based classification and regression datasets from the WILDS collection [47] that originate from real-world, safety-critical applications of deep learning (Section 5). In the spirit of Ovadia et al. [72], we focus on generalization capability and calibration on o.o.d. data, but consider more diverse and modern algorithms (Section 3) on realistic datasets with distribution shift. In particular, we include recent advances in variational inference such as natural gradient descent (iVON [53]) and low-rank posterior approximations (Rank-1 VI [17]). Furthermore, we use modern neural network architectures such as various ResNets [32], a DenseNet [35], and a transformer architecture [87]. **ii)** We present the first systematic evaluation of BDL for finetuning large pre-trained models, a setting that has recently gained attention in the context of BDL [79, 81]. We show that using BDL for finetuning gives a significant performance boost across a wide variety of tasks compared to standard deterministic finetuning (Section 5). **iii)** Inspired by the success of Deep Ensembles [51], we systematically evaluate the benefit of ensembling single-mode posterior approximations [4] (Section 5). **iv)** We use a signed extension of the expected calibration error (ECE) called the signed expected calibration error (sECE) that can differentiate between overconfidence and underconfidence, allowing us to better understand in which ways models are miscalibrated (Section 4). **v)** We compare the posterior approximation quality of the considered algorithms using the HMC samples from Izmailov et al. [37] (Section 5) and show that modern single-mode BDL algorithms approximate the parameter posterior better than Deep Ensembles, with further gains being achieved by ensembling these algorithms. Overall, our work is similar in spirit to Band et al. [4], but we compare the algorithms on more diverse datasets and focus on pure calibration metrics, thereby revealing failure modes of SOTA BDL algorithms that are not yet present in the literature. We provide code for all implemented algorithms and all evaluations[3].

## 2 Related Work

Several recent publications [1, 25] review the SOTA in uncertainty quantification using Bayesian models without providing experimental results. Yao et al. [93] compare a wide range of Markov Chain Monte Carlo [31] and approximate inference [6] methods on toy classification and regression datasets. Ovadia et al. [72] perform a large-scale experimental evaluation of a small selection of popular BDL algorithms on o.o.d. data and conclude that Deep Ensembles [51] perform best while stochastic variational inference [26] performs worst. Filos et al. [19] use a similar selection of algorithms but only evaluate on a single, large computer vision task not considering o.o.d. data. Foong et al. [20] artificially create o.o.d. splits for UCI datasets [16] and again find that variational inference performs worse than the Laplace approximation [57]. Mukhoti and Gal [63] and Gustafsson et al. [29] compare Monte Carlo Dropout [22] and Deep Ensembles in the context of semantic segmentation and depth completion, but, again, do not consider o.o.d. data. Nado et al. [65] evaluate many popular BDL algorithms on a small number of mostly artificially created o.o.d. image and text classification tasks. The work of Band et al. [4] is the most similar to ours, as they evaluate several BDL algorithms, including ensembles of single-mode posterior approximations, on two large image-classification datasets and consider o.o.d. data. Compared to Band et al. [4], we evaluate a different set of algorithms such as SWAG [59] and natural gradient descent variational inference [53]

[2]https://nips.cc/Conferences/2021/Schedule?showEvent=21827
[3]https://github.com/Feuermagier/Beyond_Deep_Ensembles

on a more diverse selection of datasets and network architectures, including transformer-based models and finetuning tasks, thereby revealing new failure modes of SOTA BDL methods. Competitions such as Wilson et al. [90] and Malinin et al. [60] also provide insights into the performance of different algorithms. However, the employed algorithms are typically highly tuned and modified for the specific tasks and thus of limited use to assess the general quality of the underlying methods in more diverse settings. Importantly, all winners of Wilson et al. [90] use ensemble-based algorithms.

# 3 Bayesian Deep Learning Algorithms

We assume a neural network with parameters $\boldsymbol{\theta}$ that models the likelihood $p(\boldsymbol{y} \mid \boldsymbol{x}, \boldsymbol{\theta})$ of an output $\boldsymbol{y}$ given an input $\boldsymbol{x}$. Treating $\boldsymbol{\theta}$ as a random variable, the parameter posterior $p(\boldsymbol{\theta} \mid \mathcal{D})$ given a training dataset $\mathcal{D} = \{(\boldsymbol{x}_i, \boldsymbol{y}_i) \mid i = 1, \ldots, N\}$ of input-output pairs is defined by Bayes' theorem as

$$p(\boldsymbol{\theta} \mid \mathcal{D}) = \frac{\prod_i p(\boldsymbol{y}_i \mid \boldsymbol{x}_i, \boldsymbol{\theta})\, p(\boldsymbol{\theta})}{\int \prod_i p(\boldsymbol{y}_i \mid \boldsymbol{x}_i, \boldsymbol{\theta})\, p(\boldsymbol{\theta})\, \mathrm{d}\boldsymbol{\theta}}, \tag{1}$$

where $p(\boldsymbol{\theta})$ is a prior over parameters. The posterior $p(\boldsymbol{\theta} \mid \mathcal{D})$ assigns higher probability to parameter vectors that fit the training data well and conform to our prior beliefs. Using $p(\boldsymbol{\theta} \mid \mathcal{D})$, a prediction $\boldsymbol{y}$ given an input vector $\boldsymbol{x}$ is defined as

$$p(\boldsymbol{y} \mid \boldsymbol{x}, \mathcal{D}) = \int p(\boldsymbol{y} \mid \boldsymbol{x}, \boldsymbol{\theta})\, p(\boldsymbol{\theta} \mid \mathcal{D})\, \mathrm{d}\boldsymbol{\theta} = \mathop{\mathbb{E}}_{\boldsymbol{\theta} \sim p(\boldsymbol{\theta}|\mathcal{D})} [p(\boldsymbol{y} \mid \boldsymbol{x}, \boldsymbol{\theta})]. \tag{2}$$

This so-called Bayesian model average (BMA) [6, 58, 64, 89] encompasses the information of all explanations of the training data that are consistent with the parameter posterior. The BMA is especially valuable when dealing with large neural networks that are typically underspecified by the training data, where marginalizing over parameters can mitigate overfitting and promises significant accuracy and calibration gains [89]. While recent work has shown problems with certain types of covariate shift [36] and o.o.d. data [11], BDL not only promises calibration but also generalization gains [89].

## 3.1 Scalable Approximations for Bayesian Deep Learning

As computing the marginalization integral defining the normalization constant of the parameter posterior (Equation (1)) is intractable for neural networks, we have to resort to approximations. Approximate inference algorithms approximate the posterior, either by sampling from it or by computing an approximate distribution. Sampling-based Markov Chain Monte Carlo (MCMC) methods [31] such as Hamiltonian Monte Carlo (HMC) [67] sample directly from the true posterior and are therefore asymptotically exact. However, they are computationally very expensive and hence typically intractable in the context of BDL. Deterministic methods such as variational inference construct local approximations at a mode of the parameter posterior and are generally more computationally performant than MCMC [37], as they transform the posterior inference problem into an optimization problem that can be efficiently solved with standard gradient-based optimization techniques [6, 58, 64]. Therefore, we focus on these algorithms

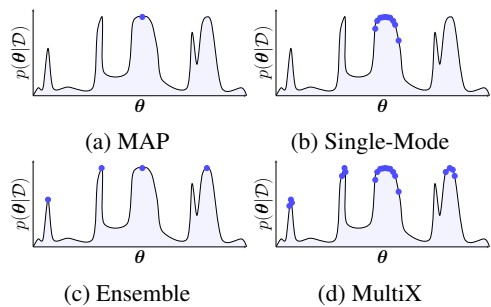

(a) MAP      (b) Single-Mode

(c) Ensemble      (d) MultiX

Figure 1: Posterior Approximation Types. MAP approximates a single posterior mode with a point estimate, while probabilistic single-mode approximations additionally capture the shape of the mode. Deep Ensembles approximate multiple modes with a mixture of point estimates. Likewise, MultiX employs a mixture of single-mode approximations to capture the shape of multiple modes. Figure adapted from Wilson and Izmailov [89].

in this work. This framework also encompasses standard deep learning, which is equivalent to a "Maximum A Posteriori" (MAP) estimate, i.e., a point estimate at the posterior maximum. In this section, we give a brief overview of the algorithms that we evaluate. See Appendix A for more detailed explanations and Appendix D for implementation details.

**Variational Inference.** Variational inference (VI) minimizes the Kullback-Leibler divergence [50] between the approximate posterior and the true posterior [26]. Bayes By Backprop (**BBB**) [7] approximates the posterior with a diagonal Gaussian distribution and optimizes the mean and variance parameters with Stochastic Gradient Descent (SGD) [42]. Rank-1 variational inference (**Rank-1 VI**) [17] in contrast uses a low-rank posterior approximation, which reduces the number of additional parameters and allows the use of multiple components in the low-rank subspace. The improved Variational Online Newton (**iVON**) algorithm [53] still uses a diagonal Gaussian posterior but uses second-order information to better optimize the distribution parameters with natural gradients. Stein Variational Gradient Descent (**SVGD**) [55] is a non-parametric VI algorithm that approximates the posterior with multiple point estimates. SVGD is similar to a Deep Ensemble (see below) but adds repulsive forces between the particles to push them away from each other in parameter space.

**Other Algorithms.** Lakshminarayanan et al. [51] introduce **Deep Ensembles** that approximate the posterior with a few, typically five to ten, independently trained MAP models. As such, Deep Ensembles were originally considered a competing approach to Bayesian models [51] but can be viewed as Bayesian as they form a sum of delta distributions that approximate the posterior [89]. We follow this interpretation. The **Laplace** approximation [57] approximates the posterior with a second-order Taylor expansion around the parameters of a MAP model. We only consider the last-layer Laplace approximation [13] with diagonal and Kronecker-factorized [74] posterior approximations, which Daxberger et al. [13] find to achieve the best tradeoff between performance and calibration. Monte Carlo Dropout (**MCD**) [22] utilizes the probabilistic nature of dropout units that are part of many common network architectures to construct an approximation of a posterior mode. Stochastic Weight Averaging-Gaussian (**SWAG**) [59] periodically stores the parameters during SGD training and uses them to build a low-rank Gaussian posterior approximation. Finally, we evaluate Spectrally-Normalized Gaussian Processes (**SNGP**) [54] as a Bayesian baseline that does not infer a distribution over the model's parameters, but replaces the last layer by a Gaussian Process. The results for SNGP are therefore not directly comparable to the other algorithms' results.

## 3.2 MultiX

While single-mode posterior approximations such as BBB and SWAG capture the shape of a single mode of the parameter posterior, Deep Ensembles cover multiple modes but approximate each with a single point estimate. Hence, ensembling single-mode approximations promises even better posterior coverage and therefore improved uncertainty estimates (see Figure 1). This concept is not new: Tomczak et al. [86] experiment with an ensemble of BBB models on small datasets. Cobb et al. [10] use an ensemble of Concrete Dropout [23] models and Filos et al. [19] use MCD models. Both report accuracy improvements compared to a Deep Ensemble. Wilson and Izmailov [89] introduce MultiSWAG, an ensemble of SWAG [59] models. The winning teams of Wilson et al. [90] also show that ensembling Bayesian neural networks yields good posterior approximations. Similar to Band et al. [4] and Mehrtens et al. [61], we ensemble all considered single-mode posterior approximations (Section 3.1) except for SNGP to assess the performance gains on a per-algorithm basis. We use the term "MultiX" to refer to an ensemble of models trained with algorithm "X". We make an exception for "MultiMAP", which we keep referring to as Deep Ensemble for consistency with the existing literature.

## 4 Calibration Metrics

A calibrated model is defined as a model that makes confident predictions if and only if they will likely be accurate. While this definition directly implies a calibration metric for classification tasks [66], it has to be adapted for regression tasks, as "being accurate" is not a binary property in the regression case.

### 4.1 Unsigned Calibration metrics

**Calibrated Classification.** The calibration of a *classification* model can be measured with the expected calibration error (ECE) [28, 66]. By partitioning the interval $[0, 1]$ into $M$ equally spaced bins and grouping the model's predictions into those bins based on their confidence values, we can calculate the average accuracy and confidence of each bin. The expected calibration error is then

given by ECE $= \sum_{m=1}^{M} |B_m|/|\mathcal{D}'||\text{acc}(B_m) - \text{conf}(B_m)|$ where $B_m$ is the set of predictions in the $m$-th bin, and $\text{acc}(B_m)$ and $\text{conf}(B_m)$ are the average accuracy and confidence of the predictions in $B_m$ (see Appendix B for details). An ECE of zero indicates perfect calibration.

**Calibrated Regression.** The confidence intervals of the predictive distribution can be used to measure the calibration of a *regression* model. Selecting $M$ confidence levels $\rho_m$ allows the computation of a calibration error based on the observed probability $p_{\text{obs}}(\rho_m)$, calculated as the fraction of predictions that fall into the $\rho_m$-confidence interval of their respective predictive distributions: QCE $= 1/M \sum_{m=1}^{M} |p_{\text{obs}}(\rho_m) - \rho_m|$. We refer to this as the quantile calibration error (QCE), which simply replaces the quantiles in the definition of the calibration error from Kuleshov et al. [49] by confidence intervals. Using the confidence intervals allows a simpler interpretation of the resulting reliability diagrams (see Appendix B).

## 4.2 Signed Calibration Metrics

Models can be miscalibrated in two distinct ways: Overconfident models make inaccurate predictions with high confidence, and underconfident models make accurate predictions with low confidence. Arguably, overconfidence is worse in practice when applicants want to rely on the model's confidence to assess whether they can trust a prediction, for example in safety-critical applications of deep learning. However, none of the presented metrics can differentiate between overconfidence and underconfidence. Until now, this information was only apparent in reliability diagrams [28]. We propose two simple extensions of the ECE and the QCE that condense the information about overconfidence and underconfidence into a single scalar value by removing the absolute values: sECE and sQCE. We define these signed calibration metrics as

$$\text{sECE} = \sum_{m=1}^{M} \frac{|B_m|}{|\mathcal{D}'|} \big(\text{acc}(B_m) - \text{conf}(B_m)\big) \quad \text{and} \quad \text{sQCE} = \frac{1}{M} \sum_{m=1}^{M} \big(p_{\text{obs}}(\rho_m) - \rho_m\big). \quad (3)$$

A positive signed calibration error indicates that a model makes predominantly underconfident predictions and a negative signed calibration error indicates predominantly overconfident predictions. Perfectly calibrated models have a sECE/sQCE of zero. For models that are overconfident for some inputs but underconfident for others the signed calibration metrics may be zero, even though the model is not perfectly calibrated. This is typically not an issue in practice, as our experiments in Appendix C show that the absolute value of the signed metrics is usually very close to the absolute value of the corresponding unsigned metric, as most models are either overconfident or underconfident for nearly all predictions. Nevertheless, we always report the signed calibration metrics together with the unsigned calibration metrics to avoid any ambiguity.

## 5 Empirical Evaluation

For our comparison of the BDL algorithms introduced in Section 3.1, we focus on **i)** the ability of the models to generalize to realistic distribution-shifted data, **ii)** the calibration of the models under distribution shift, and **iii)** how well the models approximate the true parameter posterior. To assess the generalization capability and calibration of the models under realistic distribution shift, we use a subset of the WILDS dataset collection [47]. We assess the posterior approximation quality by comparing the model's predictive distributions to those of the HMC approximation provided by Izmailov et al. [37] for CIFAR-10 [48]. See Section 5.1 and Section 5.2 for details about the task, Section 5.3 for generalization results, Section 5.4 for calibration results, and Section 5.5 for posterior approximation quality results. All results for WILDS are directly comparable to the respective non-BDL o.o.d. detection algorithms on the WILDS leaderboard[4], since we strictly follow their training and evaluation protocol. Whenever the BDL models' accuracy is competitive with the best performing algorithm on the WILDS leaderboard, the WILDS result is marked in the respective plot. We also report results for a subset of the smaller UCI [16] and UCI-Gap [20] tabular regression datasets in Appendix G.1. Appendix F contains information about the used computational resources and training times. Details regarding hyperparameters, training procedures, and additional results can be found in Appendix G. The results on all datasets are reported with a $95\%$ confidence interval.

---

[4] https://wilds.stanford.edu/leaderboard/, last accessed on August 31, 2023

## 5.1 The WILDS Datasets

WILDS consists of ten diverse datasets that originate from real-world applications of deep learning in which models need to perform well under distribution shift. Standard o.o.d. datasets such as MNIST-C [62], CIFAR-10-C [33] and UCI-Gap [20] create distribution-shifted data by selectively removing data from the training split or artificially adding data corruptions onto the data in the evaluation split. WILDS represents real-world distribution shifts and is therefore more suitable for an application-oriented evaluation of BDL. We systematically evaluate all considered algorithms (see Section 3.1) on six of the ten datasets: The image-based regression task POVERTYMAP, the image classification tasks IWILDCAM, FMOW, and RXRX1, and the text classification tasks CIVILCOMMENTS and AMAZON. Aside from POVERTYMAP, all datasets are finetuning tasks, where we initialize the model's parameters from a model that has been pre-trained on a similar task. We also evaluate some algorithms on the CAMELYON17 image classification dataset but find that the performance degradation on the o.o.d. evaluation split is to a large part a consequence of the use of batch normalization rather than the o.o.d. data, making the dataset less interesting for a fair comparison on o.o.d. data (see Appendix E for details).

As we want to evaluate the posterior approximation, generalization, and calibration capability of all models given the true parameter posterior, none of our models use the metadata (e.g. location, time) associated with the input data, nor do we consider approaches that are specifically designed for o.o.d. generalization or augment the dataset, for example by re-weighting underrepresented classes, contrary to the algorithms evaluated by Koh et al. [47].

**Large-Scale Regression.** **POVERTYMAP-WILDS** [94] is an image-based regression task, where the goal is to better target humanitarian aid in Africa by estimating the asset wealth index of an area using satellite images. As the task is significantly easier when buildings are visible in the images, the evaluation set is split into images containing urban and images containing rural areas. The accuracy of the models is evaluated on both splits by the Pearson coefficient between their predictions and the ground truth, and the worst Pearson coefficient is used as the main evaluation metric. All models are based on a ResNet-18 [32]. See Figure 3 for the Pearson coefficient and sQCE on the o.o.d. evaluation split and Appendix G.3.2 for further details.

**Finetuning of CNNs.** **IWILDCAM-WILDS** [5] is an image classification task that consists of animal photos taken by camera traps across the world. The model's task is to determine which of 182 animal species can be seen in the image. As rare animal species, which are of special interest to researchers, are naturally underrepresented in the dataset, the macro F1 score is used to evaluate the predictive performance. The o.o.d. evaluation split consists of images from new camera locations. All models are based on a ResNet-50 [32]. See Figure 2a for the macro F1 score and sECE on the o.o.d. evaluation split and Appendix G.3.3 for further details. **FMOW-WILDS** (Functional Map of the World) [9] is an image classification task, where the inputs are satellite images and the class is one of 62 building and land use categories. The o.o.d. evaluation split consists of images from different years than the images in the training set. Models are separately evaluated on five geographical regions of the world, with the lowest accuracy taken as the main evaluation metric. All models are based on a DenseNet-121 [35]. See Figure 2b for the accuracy and sECE for the region of the o.o.d. evaluation split the models perform worst on and Appendix G.3.4 for further details. **RXRX1-WILDS** [85] is an image classification task, where the inputs are three-channel images of cells, and the classes are 1139 applied genetic treatments. The o.o.d. evaluation split is formed by images from different experimental batches than the training data. Following Koh et al. [47], we only use three of the six available input channels to limit the computational complexity of the models. This makes the task considerably harder and leads to the low accuracy of the models, but makes our results comparable to those of Koh et al. [47]. All models are based on a ResNet-50 [32]. See Figure 2c for the accuracy and sECE on the o.o.d. evaluation split and Appendix G.3.5 for further details.

**Finetuning of Transformers.** **CIVILCOMMENTS-WILDS** [8] is a binary text classification dataset, where the model's task is to classify whether a given comment is toxic or not. The comments are grouped based on whether they mention certain demographic groups, such as LGBTQ or Muslim identities. Models are evaluated based on the group on which they achieve the lowest accuracy on the evaluation set. All models are based on the DistilBERT architecture [76]. See Figure 4a for the accuracy and sECE on the group of the o.o.d. evaluation split the models perform worst on and Appendix G.3.6 for further details. **AMAZON-WILDS** [68] consists of textual product reviews, where

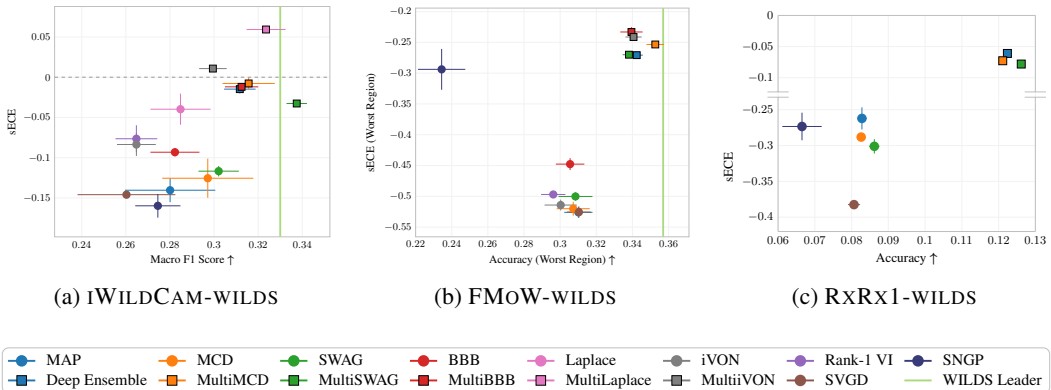

(a) IWILDCAM-WILDS      (b) FMOW-WILDS      (c) RXRX1-WILDS

Figure 2: Accuracy Metrics vs. sECE on the o.o.d. evaluation splits of the image classification finetuning tasks IWILDCAM-WILDS, FMOW-WILDS, and RXRX1-WILDS. Note the split y-axis in Figure 2c. All MultiX algorithms are more accurate and better calibrated than any single-mode approximation. Except for RXRX1, BBB is better calibrated than MCD and SWAG. iVON's calibration is inconsistent: On IWILDCAM it is better calibrated than BBB, but on FMOW it barely performs better than MAP. On RXRX1, the VI algorithms except for SVGD are significantly less accurate than all other models. We experimented with different hyperparameters in Appendix G.3.5. Laplace is very well calibrated on IWILDCAM, but underperforms on FMOW and RXRX1. SVGD performs very similarly to MAP regarding both metrics, even though it uses a multi-mode posterior approximation.

the task is to predict the star rating from one to five. The o.o.d. evaluation split consists of reviews from reviewers that are not part of the training split. Models are evaluated based on the accuracy of the reviewer at the $10\%$ quantile. All models are based on DistilBERT [76]. See Figure 4b for the accuracy and sECE on the o.o.d. evaluation split and Appendix G.3.7 for further details.

## 5.2 The Corrupted CIFAR-10 Dataset

CIFAR-10-C [33] is a corrupted version of the evaluation split of the image classification dataset CIFAR-10 [48], where images are corrupted with increasing levels of noise, blur, and weather and digital artifacts. We compare the considered algorithms on the standard evaluation split of CIFAR-10 as well as the corruption levels 1, 3, and 5 of CIFAR-10-C. Following Izmailov et al. [37], all of our models on CIFAR-10-(C) are based on the ResNet-20 architecture. See Appendix G.3.6 for details.

## 5.3 Generalization to Realistic Distribution Shift

We measure the generalization capability of the models with the task-specific accuracy metrics proposed by Koh et al. [47] that are based on the real-world origin of the respective tasks. The metrics typically emphasize the performance on groups or classes that are underrepresented in the training data, as avoiding bias against these groups is crucial in safety-critical applications of BDL.

Except for the text classification tasks, MultiX always generalizes better than single-mode posterior approximations. Overall, the relative ordering of the MultiX models depends on the dataset and in many cases does not correlate with the relative ordering of the corresponding single-mode approximations.

**Large-Scale Regression.** All models achieve similar Pearson coefficients, with MultiX being slightly more accurate. The Deep Ensemble is competitive with the best performing algorithm of the WILDS leaderboard with a Pearson coefficient of $0.52$ compared to $0.53$ of C-Mixup [92]. However, due to the large standard errors resulting from the different difficulties of the folds, the results are not significant. Note that Koh et al. [47] report similarly large standard errors.

**Finetuning of CNNs.** Confirming the overall trend, MultiX models generalize better than single-mode models, with MultiSWAG and MultiMCD performing particularly well. Except for RXRX1 the

models perform competitively with the best models from the WILDS leaderboard. On IWILDCAM, the single-mode posterior approximations SWAG and MCD are competitive with the Deep Ensemble. SVGD performs similarly to MAP, even though it is based on an ensemble, likely due to the repulsive forces pushing the particles away from the well-performing pre-trained model. While the VI algorithms' accuracy is similar to the accuracy of MAP on IWILDCAM and FMoW, all VI algorithms except SVGD perform significantly worse than the non-VI algorithms on RXRX1. Laplace is well calibrated on IWILDCAM, but significantly less accurate than the other algorithms on FMoW and RXRX1. This seems to represent a fundamental approximation failure of Laplace, and not only a sampling issue, since increasing the number of samples lead to only a small increase in accuracy (see Appendix G.3.4 and Appendix G.3.5).

**Finetuning of Transformers.** BBB and Rank-1 VI are the most accurate models on both tasks, with no benefit from the multiple components of Rank-1 VI. Interestingly, iVON is significantly less accurate than BBB, even though it is also based on mean-field VI, indicating that the natural gradient-based training is disadvantageous on the transformer-based BERT architecture. To see whether the better performance of BBB is due to less regularization compared to MAP, we also experiment with a smaller weight decay factor for MAP on CIVILCOMMENTS. While we find that the accuracy increases, BBB is still more accurate (see Appendix G.3.6). Finally, we also check whether the better performance of BBB is due to its last-layer nature. We experiment with last-layer versions of MCD and SWAG on AMAZON (see Appendix G.3.7), but find that both are still significantly less accurate than BBB.

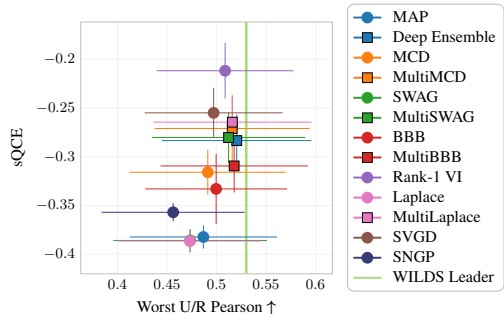

Figure 3: POVERTYMAP-WILDS: Worst urban/rural Pearson coefficient between the model's predictions and the ground truth plotted against the sQCE on the o.o.d. test split of the image-based regression task. All models achieve similar, but noisy [47], Pearson coefficients, indicating similar generalization capabilities. Multi-mode approximations are consistently better calibrated than single-mode approximations (note that Rank-1 VI's components and SVGD's particles give them multi-mode approximation capabilities). Regarding calibration, the relative ordering of the single-mode models does not translate to the MultiX models: BBB is among the best-calibrated single-mode models, but MultiBBB is the worst calibrated MultiX model. Laplace and SWAG are very similarly calibrated, therefore the data points of SWAG are hidden behind the data points of Laplace. iVON performs significantly worse than the other algorithms and is therefore excluded.

MultiX is no more accurate than the corresponding single-mode approximation, contrary to the results on all other datasets. We suspect that this effect is to a large part due to the finetuning nature of the tasks, where all ensemble members start close to each other in parameter space and therefore converge to the same posterior mode. Note that the failure of ensembles is most likely due to the task and the network architecture and not due to the training procedure: While we train for fewer epochs than on the image classification tasks, the datasets are larger. On IWILDCAM we perform 97k parameter updates, compared to 84k parameter updates on CIVILCOMMENTS.

## 5.4 Calibration under Realistic Distribution Shift

We measure calibration with the sECE for classification tasks and with the sQCE for regression tasks (see Section 4). We additionally report the unsigned ECE/QCE and the log-likelihood for the regression task in Appendix G.

MultiX is almost always less overconfident than single-mode approximations. When all models are already comparatively well calibrated, MultiX tends to become underconfident. Thus, we find that MultiX is typically only less confident, but not automatically better calibrated than single-mode approximations. On the transformer-based text classification tasks, MultiX is almost never better calibrated than the respective single-mode approximation.

**Large-Scale Regression.** MultiX, when based on a probabilistic single-mode approximation, is generally better calibrated than the Deep Ensemble. SVGD is better calibrated than the Deep Ensemble, showing the benefit of the repulsive forces between the particles. Rank-1 VI is the best calibrated model, indicating that multi-modality over all parameters as with the Deep Ensemble is not necessary to capture the multi-modality of the parameter posterior.

**Finetuning of CNNs.** Again, we find that MultiX generally performs better than a Deep Ensemble. However, MultiSWAG in particular is more overconfident than the Deep Ensemble, even though SWAG is better calibrated than MAP. BBB is better calibrated than other single-mode approximations such as SWAG and MCD. Rank-1 VI performs similar to BBB on all tasks, indicating that the low-rank components are not sufficient to capture the multi-modality of the parameter posterior in the finetuning setting. On IWILDCAM, Laplace is the best calibrated single-mode approximation, and correspondingly Multi-Laplace is the most underconfident multi-mode approximation. This result is unique to IWILDCAM, as Laplace tends to be overconfident on the other image classification datasets.

**Finetuning of Transformers.** Except for MultiBBB on AMAZON, ensembles are similarly calibrated than the respective single-mode approximations. SWAG is the least confident model on both tasks, which leads to underconfidence on AMAZON. MCD's calibration is inconclusive, as it is better calibrated than MAP on AMAZON, but more overconfident on CIVILCOMMENTS. BBB and Rank-1 VI are not better calibrated than MAP and on AMAZON significantly more overconfident than MAP.

## 5.5 Posterior Approximation Quality

While approximate inference is commonplace in BDL, the large size of the neural networks typically makes it computationally intractable to measure how well a model approximates the true parameter posterior. Following Wilson et al. [90] and using the HMC samples provided by Izmailov et al. [37], we measure how well the models approximate the predictive distribution of HMC by the total variation (TV) between the model's predictions and HMC and the top-1 agreement with HMC on CIFAR-10-(C) [33, 48]. While the TV in the predictive space is indicative of the parameter posterior approximation quality, it does not allow for definite conclusions about the parameter space. Figure 5 displays the TV of the evaluated models under increasing levels of image corruption. For further results regarding the accuracy, sECE, ECE, and top-1 agreement with HMC see Appendix G.2.

Overall, a good probabilistic single-mode approximation is the most important factor for a good posterior approximation. MultiX, when based on probabilistic single-mode approximations, consistently approximates the parameter posterior better than single-mode-only approximations and the Deep Ensemble. MultiiVON approxi-

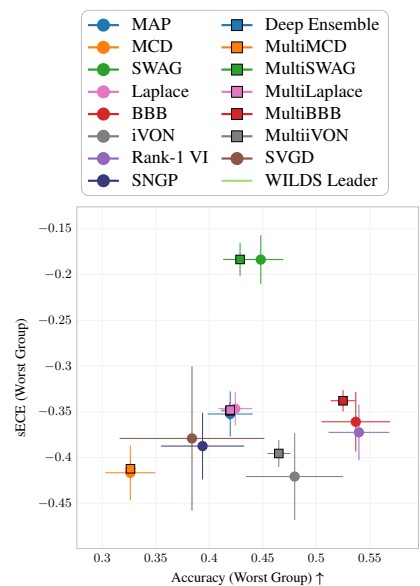

(a) CIVILCOMMENTS-WILDS

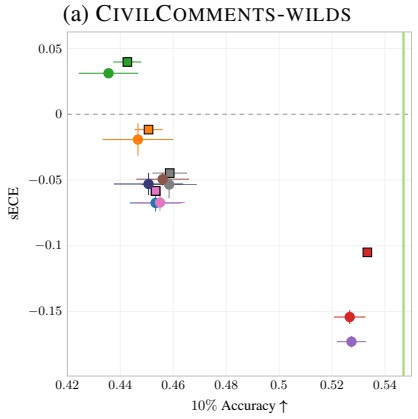

(b) AMAZON-WILDS

Figure 4: Text classification with pre-trained transformers. Except for Multi-BBB on AMAZON-WILDS, MultiX performs nearly identically to the corresponding single-mode approximation. VI improves the accuracy of the models. MCD is the least accurate model on CIVILCOMMENTS. We experiment with different dropout rates in Appendix G.3.6 but find that MCD never outperforms MAP.

mates the posterior best across all corruption levels as measured by the TV, with MultiSWAG being a close contender. Even single-mode approximations such as MCD and SWAG achieve better TVs

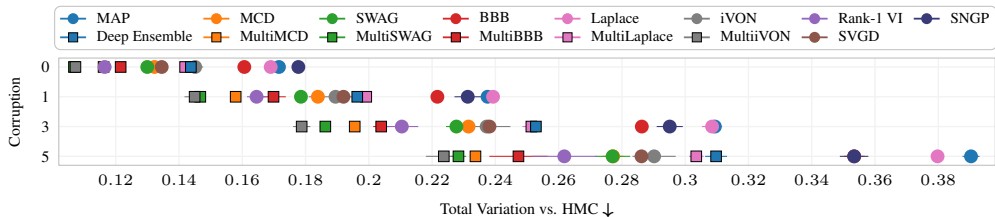

Figure 5: CIFAR-10-(C): Total variation (TV, smaller is better) between the model's predictions and HMC's predictions on the evaluation split of CIFAR-10 (corruption 0) and under increasing levels of image corruption by noise, blur, and weather and digital artifacts (corruption 1, 3, 5). MultiX consistently approximates the posterior better than the corresponding single-mode approximations, with MultiiVON and MultiSWAG achieving the smallest TV. The single-mode approximations SWAG, MCD, and iVON approximate the posterior better than the Deep Ensemble.

under data corruption than the Deep Ensemble. As expected, MAP has the highest TV, with only a small improvement made by Laplace.

## 6 Conclusion

We presented a comprehensive evaluation of a wide range of modern, scalable BDL algorithms, using distribution-shifted data based on real-world applications of deep learning. We focused on the generalization capability, calibration, and posterior approximation quality under distribution shift. Overall, our analysis resulted in the following takeaway messages:

1. Finetuning only the last layers of pre-trained models with BDL algorithms gives a significant boost of generalization accuracy and calibration on realistic distribution-shifted data, while incurring a comparatively small runtime overhead. These models are in many cases competitive to or even outperform methods that are specially designed for OOD generalization such as IRM [3] and Fish [80].

2. For CNNs, ensembles are more accurate and better calibrated on OOD data than single-mode posterior approximations by a wide margin, even when initializing all ensemble members from the same pre-trained checkpoint with only the last layers differently initialized, i.e. when not using the standard protocol of randomly initializing all ensemble members. Ensembling probabilistic single-mode posterior approximations such as SWAG or MCD yields only a small additional increase in accuracy and calibration.

3. When finetuning large transformers, ensembles, which are typically considered to be the SOTA in BDL, yield no benefit. Compared to all other evaluated BDL algorithms, classical mean-field variational inference achieves significant accuracy gains under distribution shift.

**Limitations.** While we evaluate on a wide range of datasets from different domains and using different network architectures, the choice of tasks is still limited. In particular, we do not consider LSTMs [34] as Ovadia et al. [72] do. Given the limitations of WILDS [47], we evaluate on a single large-scale regression dataset. As both text classification experiments use DistilBERT [76], it is conceivable that the failure of ensembles is limited to this particular architecture. We do not include algorithms that are based on function-space priors [30, 56, 75, 84]. Except for the results on CIFAR-10-(C) and POVERTYMAP, all results were obtained by finetuning pre-trained models and are therefore only valid in this setting. The HMC samples used in Section 5.5 have been criticized for not faithfully representing the true parameter posterior due to low agreement between the predictions of different chains [78].

**Broader Context.** Bayesian deep learning aims to provide reasonable uncertainty estimates in safety-critical applications. Hence, we do not expect any societal harm from our work, as long as it is ensured by proper evaluation that accuracy and calibration requirements are met before deployment.

## Acknowledgments

This work was supported by funding from the pilot program Core Informatics of the Helmholtz Association (HGF). The authors acknowledge support by the state of Baden-Württemberg through bwHPC, as well as the HoreKa supercomputer funded by the Ministry of Science, Research and the Arts Baden-Württemberg and by the German Federal Ministry of Education and Research.

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

# A  Approximate Inference

This section provides further details on the algorithms introduced in Section 3.1.

## A.1  Variational Inference

Variational inference (VI) minimizes the KL divergence [50] between the true posterior $p(\boldsymbol{\theta} \mid \mathcal{D})$ and the approximate posterior $q(\boldsymbol{\theta} \mid \mathcal{D})$ [6]. While the KL divergence cannot be computed by itself, as the true posterior is unknown, it can still be minimized by maximizing the evidence lower bound (ELBO) given the parameter prior $p(\boldsymbol{\theta})$:

$$\text{ELBO} = \mathop{\mathbb{E}}_{\boldsymbol{\theta} \sim q(\boldsymbol{\theta}|\mathcal{D})} \left[ \log p(\mathcal{D} \mid \boldsymbol{\theta}) \right] - \text{KL} \left[ q(\boldsymbol{\theta} \mid \mathcal{D}) \mid\mid p(\boldsymbol{\theta}) \right] \tag{4}$$

Maximizing the ELBO means maximizing the likelihood of the training data, therefore fitting the data well, while staying close to the parameter prior [6].

**Bayes By Backprop (BBB).**  BBB [7] is an application of VI to deep neural network. BBB approximates the parameter posterior with a diagonal Gaussian distribution that cannot model covariances between parameters. The per-parameter means and variances are learned with standard Stochastic Gradient Descent (SGD) [42] using the negative of the ELBO as the loss function. The ELBO by itself is not differentiable as it depends on the randomly chosen parameters. However, the reparameterization trick [44] applies to diagonal Gaussians and allows us to use the negative ELBO as the loss function. Further runtime performance improvements are possible by using the local reparameterization trick [45] or Flipout [88].

While there have been reports of BBB performing well when used on neural networks [7, 86], the current consensus of the research community seems to be that BBB falls short when compared to e.g. ensembles [20, 72, 89], even though it has been shown that the diagonal Gaussian posterior is not significantly less expressive than a posterior that models covariances [18, 20]. In recent years significant work has been done to improve the performance of VI in a deep learning setting. To assess whether these improved algorithms can compete with SOTA Bayesian algorithms, we also evaluate promising improvements on posterior parameterizations (Rank-1 VI, SVGD) and optimization procedures (iVON).

**Rank-1 Variational Inference (Rank-1 VI).**  Rank-1 VI [17] enhances the posterior approximation of BBB by approximating a full-rank covariance matrix with a low-rank approximation. Rank-1 VI learns a diagonal Gaussian distribution over two vectors per layer, whose outer product is then element-wise multiplied to a learned point estimate of the layer's weights. The bias vector is kept as a point estimate. The limited number of additional parameters allows Rank-1 VI to learn a multi-component Gaussian distribution for the two low-rank vectors, which gives Rank-1 VI ensemble-like properties. Rank-1 VI is both less expressive than BBB with the mean field approximation in the sense that it has fewer variational parameters, and is more expressive as it can model covariances between parameters within a layer and can express multi-modality in a limited way.

**Improved Variational Online Newton (iVON).**  The usage of SGD for the optimization of variational parameters is problematic, as these parameters form a complex, non-euclidean manifold [41]. Natural gradient descent (NGD), recently formalized as the Bayesian learning rule [39], exploits this structure to speed up training. VOGN [41, 70] applies NGD to neural networks but has scaling problems, as it requires per-example gradients in minibatch training. iVON, based on the improved Bayesian learning rule [53], no longer has this problem. While iVON still uses the mean-field approximation of BBB, it is expected to converge faster, and, importantly, halves the number of trainable parameters by implicitly learning per-parameter variances.

**Stein Variational Gradient Descent (SVGD).**  SVGD [55] is a non-parametric VI algorithm that does not assume the posterior to be of a particular shape but approximates it with $p$ particles (i.e. point estimates). The particles can be viewed as members of a Deep Ensemble [51], and the use of VI adds a repulsive component to the loss function based on the RBF kernel distance between the parameters of the particles. While this repulsive component can prevent the particles from converging to the same posterior mode, it prohibits the independent training of the particles.

## A.2  Other Algorithms

**Deep Ensembles.** Lakshminarayanan et al. [51] introduce Deep Ensembles that combine the predictions of multiple independently trained neural networks to improve uncertainty estimates. Originally, Deep Ensembles have been seen as a competing approach to Bayesian algorithms [51]. However, ensembles can be considered to be a Bayesian algorithm that approximates the posterior with a sum of delta distributions [89]. We consider all ensembles to be Bayesian: While they are missing the principled posterior approximation approach of VI, basically hoping that the members converge to different posterior modes, the approach results in a posterior approximation that is in many cases better than the approximation of for example BBB (Section 5, [72, 89]).

Ensembles are usually considered SOTA in uncertainty estimation [72, 89]. However, the training time scales linearly in the number of ensemble members. This makes them highly expensive in cases where training a single member is already expensive, such as with large networks, and opens the space for new, cheaper posterior approximations.

**Monte Carlo Dropout (MCD).** MCD [22] uses dropout [83] to form a Bernoulli distribution over network parameters. The dropout rates are typically not learned, but the dropout units that are present in many network architectures are simply applied during the evaluation of the model. This very cheap posterior approximation has been criticized for not being truly Bayesian [71]. Despite this criticism, it is still widely used, including in practical applications [10]. When the dropout rate is learned, MCD can be considered to implicitly perform VI [21].

**Stochastic Weight Averaging-Gaussian (SWAG).** SWAG [59] forms its posterior approximations from the parameter vectors that are traversed during the training of a standard neural network. During the last epochs of SGD training, SWAG periodically stores the current parameters of the neural network to build a low-rank Gaussian distribution over model parameters. While SWAG has only a very small performance overhead during training, storing the additional parameters requires a significant amount of additional memory, and sampling parameters from the low-rank Gaussian distribution incurs a performance overhead during evaluation.

**Laplace Approximation.** The Laplace approximation [57] builds a local posterior approximation from a second-order Taylor expansion around a MAP model. We always use the last-layer Laplace approximation and switch between a full-rank posterior, diagonal posterior, and a Kronecker-factorized posterior [74] depending on the task. In this configuration, the Laplace approximation is the only post-hoc algorithm that we consider: It can be fitted on top of an existing MAP model by performing a single pass on the training dataset.

# B  Unsigned Calibration Metrics

As mentioned in the main paper (Section 4), a calibrated model makes confident predictions if and only if they will likely be accurate. Based on this definition, we can directly derive a calibration metric for classification models: The expected calibration error (ECE) [28, 66]. In the regression case, neither "accuracy" nor "confidence" are well-defined properties of a prediction. The notion of calibration must therefore be adapted for regression tasks. In addition, the log marginal likelihood is commonly used to jointly evaluate the accuracy and the calibration in regression tasks. See Appendix G.3.2 for details.

**Calibrated Classification.** In the classification case, each data point has an associated distribution $Y$ over the possible labels. $Y$ represents the inherent aleatoric uncertainty of the label. Given a prediction $\hat{y} = \arg\max_y p(y \mid \boldsymbol{x}, \mathcal{D})$ made with confidence $\hat{p} = \max_y p(y \mid \boldsymbol{x}, \mathcal{D})$, the model is perfectly calibrated if and only if

$$\mathbb{P}\left(\hat{y} = Y \mid \hat{p} = p\right) = p \qquad \forall p \in [0, 1] \tag{5}$$

holds for every data point [12, 28, 66]. Informally speaking, this means that if the model makes 100 predictions with a confidence of 0.8, 80 of these predictions should be correct. The expected difference between the left and the right side of Equation (5) is called the expected calibration error (ECE) of the model:

$$\mathrm{ECE} = \mathop{\mathbb{E}}_{p \sim \mathcal{U}([0,1])} \left[\, \left|\mathbb{P}\left(\hat{y} = Y \mid \hat{p} = p\right) - p\right| \,\right] \tag{6}$$

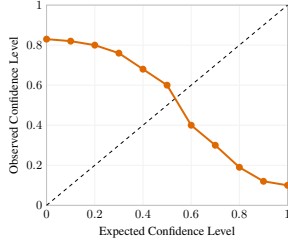
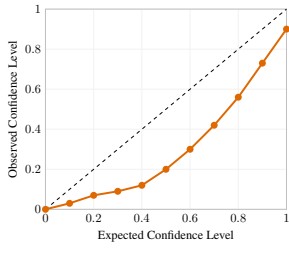

(a) Overconfident Quantile

(b) Overconfident CI

Figure 6: Reliability plots of fictional, overconfident regression models when using (a) quantiles and (b) confidence intervals (CI).

It implies two properties of a well-calibrated model: If the accuracy is low, the confidence should also be low. This means that the model must not be overconfident in its predictions. Conversely, if the accuracy is high, the confidence should also be high, meaning that the model must not be underconfident in its predictions.

In practice, a model does not make enough predictions of the same confidence to calculate the calibration error exactly. Therefore, the model's predictions on an evaluation set $\mathcal{D}'$ are commonly grouped into $M$ equally spaced bins $B_m$ based on their confidence values, and the average accuracy and confidence of each bin are used to calculate the ECE [28, 66]:

$$\text{ECE} \approx \sum_{m=1}^{M} \frac{|B_m|}{|\mathcal{D}'|} |\text{acc}(B_m) - \text{conf}(B_m)|, \tag{7}$$

where $B_m$ is the set of predictions in the $m$-th bin, and $\text{acc}(B_m)$ and $\text{conf}(B_m)$ are the average accuracy and confidence of the predictions in $B_m$:

$$\text{acc}(B_m) = \frac{1}{|B_m|} \sum_{(\boldsymbol{x}, y) \in B_m} \mathbf{1}\big(y = \arg\max_{y'} p(y' \mid \boldsymbol{x}, \mathcal{D})\big) \tag{8}$$

$$\text{conf}(B_m) = \frac{1}{|B_m|} \sum_{(\boldsymbol{x}, y) \in B_m} \max_{y'} p(y' \mid \boldsymbol{x}, \mathcal{D}) \tag{9}$$

$$\tag{10}$$

An ECE of zero indicates perfect calibration. We always use ten bins ($M = 10$).

A main problem of the ECE is that bins with few predictions in them may exhibit a high variance [69]. Therefore, Nixon et al. [69] proposed an extension of the ECE that uses bins of adaptive width.

**Calibrated Regression.** The confidence intervals of the predictive distribution can be used to measure the calibration of a *regression* model [49]. The probability of the ground-truth output $\boldsymbol{y}$ laying inside of the $\rho$-confidence interval of the predictive distribution of the model for input $\boldsymbol{x}$ should be exactly $\rho$. Formally, we say a regression model is perfectly calibrated on an evaluation dataset $\mathcal{D}'$ if and only if

$$\mathbb{P}(Q_{\rho'}(\boldsymbol{x}) \leq \boldsymbol{y} \leq Q_{1-\rho'}(\boldsymbol{x})) = \rho \qquad \forall (\boldsymbol{x}, \boldsymbol{y}) \in \mathcal{D}' \tag{11}$$

holds for every $q$-quantile $Q_q(\boldsymbol{x})$ of the predictive distribution for input $\boldsymbol{x}$ with $\rho' = {(1-\rho)}/{2}$.

Selectively evaluating Equation (11) for $M$ confidence values $\rho_m$ allows the practical computation of a quantile calibration error (QCE) on an evaluation dataset $\mathcal{D}'$

$$\text{QCE} = \frac{1}{M} \sum_{m=1}^{M} |(\rho_m - p_{\text{obs}}(\rho_m))| \tag{12}$$

with

$$p_{\text{obs}}(\rho_m) = \frac{1}{|\mathcal{D}'|} \sum_{(\boldsymbol{x}, \boldsymbol{y}) \in \mathcal{D})} \mathbf{1}(Q_{\rho'}(\boldsymbol{x}) \leq \boldsymbol{y} \leq Q_{1-\rho'}(\boldsymbol{x})). \tag{13}$$

The QCE simply replaces the quantiles in the definition of the calibration error from Kuleshov et al. [49] by confidence intervals. Using the confidence intervals allows a simpler interpretation of the resulting reliability diagrams: With the calibration error proposed by Kuleshov et al. [49], the reliability diagram of a perfectly calibrated regression model is a horizontally mirrored version of the reliability diagram of a perfectly calibrated classification model, as there are too many ground-truth values below the lower quantiles of their predictive distributions, and too few above the higher quantiles (Figure 6a). Using confidence intervals for the reliability diagram results in a plot that can be interpreted in the same way as a reliability diagram of a classification model (Figure 6b). We always use ten equally-spaced confident levels between 0 and 1 ($M = 10$).

## C   Signed Calibration Metrics

As described in the main paper, our signed calibration metrics (sECE and sQCE) may be zero even though the model is not perfectly calibrated. However, we show that this is typically not an issue in practice, as for most models nearly all predictions are overconfident or nearly all predictions are underconfident. The reliability diagrams in Figure 7 confirm this for a representative selection of overconfident and underconfident models. We always report the unsigned calibration metrics in Appendix G in addition to the signed calibration metrics mentioned in the main paper. The unsigned metrics are in almost all cases very close to the absolute value of the signed metric, resulting in the same relative ordering of the algorithms. On the other hand, the sECE provides valuable insights into the underconfidence of some algorithms such as MultiSWAG on CIFAR-10 and SWAG on AMAZON-WILDS.

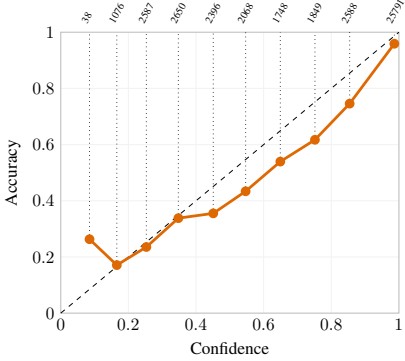

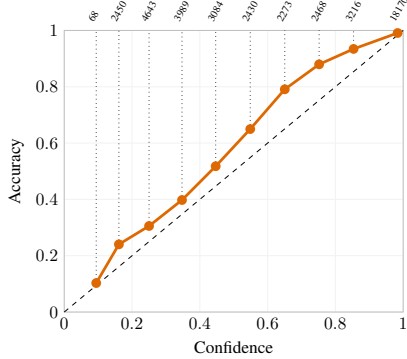

(a) MAP on the o.o.d. evaluation split of iWILDCAM-WILDS. sECE: $-0.0457$, ECE: $0.0463$

(b) MultiLaplace on the o.o.d. evaluation split of iWILDCAM-WILDS. sECE: $-0.04501$, ECE: $0.0501$

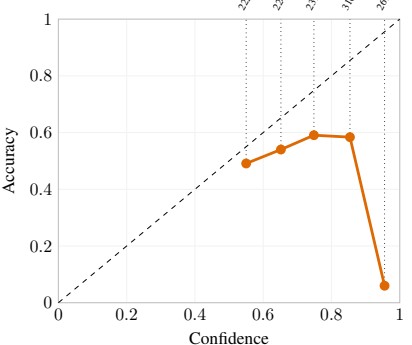

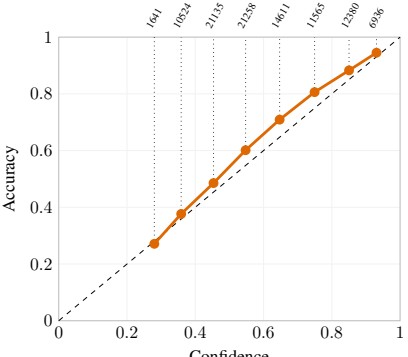

(c) MAP on the group with the worst accuracy on CIVILCOMMENTS-WILDS. sECE: $-0.3162$, ECE: $0.3162$

(d) SWAG on the o.o.d. evaluation split of AMAZON-WILDS. sECE: $-0.0405$, ECE: $0.0408$

Figure 7: Reliability diagrams of different models on a variety of datasets. No data point is drawn for empty bins. The number of predictions in each bin is denoted at the top of each plot. The dashed line corresponds to a perfectly calibrated model. In all cases, either nearly all of the model's predictions are overconfident, or nearly all are underconfident. Therefore, the ECE is close to the absolute value of the sECE, indicating that the sECE is a reasonable calibration metric.

# D   Implementation Details

Except for Laplace, we implement all algorithms ourselves as PyTorch [24] optimizers. The implementation of the algorithms as well as code to reproduce all experiments is available at https://github.com/bdl-authors/beyond-ensembles, where we also provide a short tutorial on the usage of our implementation.

**Bayes By Backprop.**   We use the local reparameterization trick [45]. As it is standard today [20, 72, 89], we do not use the scale mixture prior introduced by BBB's original authors [7], but a unit Gaussian prior. For the experiments on CIFAR-10, we make the parameters of the Filter Response Normalization layers variational.

**Rank-1 VI.**   Following Dusenberry et al. [17], we keep the bias of each layer as a point estimate. We also keep the learned parameters of batch normalization and Filter Response Normalization layers as point estimates. We use five components in most cases which is close to the four components recommended by Dusenberry et al. [17] and make Rank-1 VI directly comparable to other ensemble-based models that use five members.

**iVON.** We adapt the data augmentation factor that Osawa et al. [70] introduce for VOGN [40] to iVON. We do not use the tempering parameter from VOGN.

**Laplace.** We use the Laplace library from Daxberger et al. [13] due to the difficulty of implementing second-order optimization in PyTorch. In all cases except for CIVILCOMMENTS-WILDS, we use a Kronecker-factorized last-layer Laplace approximation. On CIVILCOMMENTS-WILDS, we use a diagonal last-layer Laplace approximation as the Kronecker-factorized approximation frequently leads to diverging parameters. We do not use the GLM approximation as proposed by Daxberger et al. [13] but use Monte Carlo sampling to stay consistent with the other evaluated algorithms. In all experiments we use the Laplace library's functions to tune the prior precision after fitting the Laplace approximation.

**SWAG.** While the authors of SWAG argue that SWAG benefits from a special learning rate schedule [59], they do not use such a schedule in most of their experiments with SWAG and MultiSWAG [89]. Correspondingly, we use the same schedule with SWAG as with any other algorithm. We use 30 parameter samples for building the mean and the low-rank covariance matrix of SWAG. On CIVILCOMMENTS-WILDS, we only use 10 parameter samples due to the storage size of the samples.

# E  Batch Normalization, Distribution Shift, and Bayesian Deep Learning

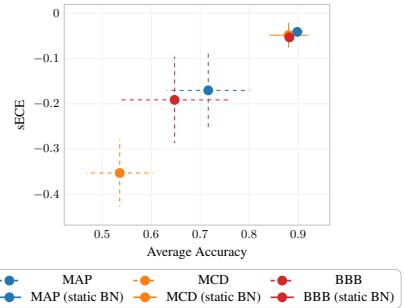

Figure 8: CAMELYON17-WILDS: Average accuracy vs. sECE on the o.o.d. test split. The models that use no running statistics (static BN) are significantly more accurate and better calibrated, while exhibiting a smaller variance.

Schneider et al. [77] find that a significant part of the accuracy loss on o.o.d. data is due to changing batch statistics that cannot be adequately normalized by the running batch normalization statistics that are based on the training data. The authors propose to re-initialize the running statistics on a subset of the evaluation dataset.

We are able to reproduce the issue with o.o.d. data on the CAMELYON17-WILDS dataset from the WILDS collection [47] (Figure 8). The o.o.d. evaluation set of CAMELYON17 has been generated by selecting the images that were most visually distinct from the other images. In addition, the employed ResNet-20 [32] architecture includes batch normalization layers. We find that using only batch statistics, thereby essentially using the batch normalization layers in training mode during evaluation, entirely alleviates the i.d. - o.o.d. performance gap on CAMELYON17, as well as the large standard deviations on the o.o.d. dataset. Coincidentally, the WILDS leaderboard [46] shows that models that do not include batch normalization, such as a model based on the vision transformer [15], or that use extensive data augmentation, perform best.

The running statistics of batch normalization layers also pose problems with Bayesian neural networks that sample parameters, as the running statistics depend on the parameters of the neural network. Wilson and Izmailov [89] therefore propose to recalculate the batch normalization statistics for each parameter sample. This is not necessary in our case as we never use running statistics for normalization layers. By doing so we also avoid the aforementioned distribution-shift problem without requiring additional o.o.d. data during evaluation, and do not add any computation overhead.

## F  Computational Resources

We use single NVIDIA Tesla V100, A100, and H100 GPUs for all tasks from Wilds [47] and CIFAR-10-(C) [33, 48]. See Table 1 for the GPUs that we use on the individual datasets as well as the runtime of MAP. Table 2 displays the relative runtime of the BDL algorithms. In total, we estimate that the evaluation required about $1600\,\text{h}$ of GPU time, of which about $25\%$ were consumed during implementation, testing and hyperparameter optimization. Training and hyperparameter optimization of the UCI models was performed on a single CPU in about $20\,\text{h}$. Table 3 shows the GPU memory overhead of the BDL algorithms.

| Dataset | GPU | Runtime of MAP |
|---|---|---|
| CIFAR-10 | NVIDIA V100 | $50\,\text{min}$ |
| POVERTYMAP-WILDS | NVIDIA V100 | $50\,\text{min}$ |
| IWILDCAM-WILDS | NVIDIA A100 | $150\,\text{min}$ |
| FMoW-WILDS | NVIDIA V100 | $150\,\text{min}$ |
| RxRx1-WILDS | NVIDIA V100 | $140\,\text{min}$ |
| CIVILCOMMENTS-WILDS | NVIDIA A100 | $60\,\text{min}$ |
| AMAZON-WILDS | NVIDIA H100 | $90\,\text{min}$ |

Table 1: Hardware and runtime for MAP for each dataset. The results are rounded to the next 10 minutes.

| Model | POVERTYMAP | IWILDCAM | FMoW | RxRx1 | CIVILCOMMENTS | AMAZON | CIFAR-10 |
|---|---|---|---|---|---|---|---|
| MAP | 1.0 | 1.0 | 1.0 | 1.0 | 1.0 | 1.0 | 1.0 |
| MCD | 1.0 | 1.0 | 1.0 | 1.0 | 1.0 | 1.0 | $\sim 1.0$ |
| SWAG | 1.3 | 1.5 | 1.0 | 1.2 | 1.3 | 1.5 | $\sim 1.0$ |
| Laplace | 1.0 | 1.0 | 1.0 | 1.0 | 1.0 | 1.0 | $\sim 1.0$ |
| BBB | 5.7 | - | - | - | - | - | $\sim 5.0$ |
| LL BBB | - | 1.6 | 2.0 | 3.7 | 1.8 | 2.0 | - |
| Rank-1 VI | 3.9 | - | - | - | - | - | $\sim 4.0$ |
| LL Rank-1 VI | - | 2.0 | 2.0 | 3.7 | 1.9 | 2.0 | - |
| iVON | 2.8 | - | - | - | - | - | $\sim 3.0$ |
| LL iVON | - | 3.0 | 2.9 | 3.6 | 5.6 | 6.6 | - |
| SVGD | 9.2 | 4.9 | 7.3 | 8.9 | 9.2 | 10.0 | $\sim 8.0$ |

Table 2: Runtime of different algorithms relative to MAP. The numbers on CIFAR-10 are conservative estimates as exact numbers were no longer available. Note that the runtime also depends on whether we are able to use mixed precision training, which was not possible with the VI algorithms. The training time of a MultiX model with $n$ members is $n$ times the training time of the respective single-mode approximation. LL = Last-Layer.

| Model | Memory Overhead |
|---|---|
| MAP | 1.0 |
| MCD | 1.0 |
| SWAG | $\sim 1.0$ |
| Laplace | $\sim 1.0$ |
| BBB | $\sim 2$ |
| Rank-1 VI | $\sim 1 + \#\text{components} \cdot \sqrt{\text{parameter count}}$ |
| iVON | $\sim 2$[5] |
| SVGD | $\sim \#\text{particles}$ |

Table 3: GPU memory requirements of different algorithms relative to MAP. The numbers are estimates are based on a theoretical analysis of the algorithms, not on measurements. The memory consumption of MultiX is the same as for the respective single-mode approximation, since all members can be trained indenpendently.

---

[5]No additional memory overhead due to a separate optimizer

# G  Additional Experimental Results

## G.1  UCI Datasets

We report results for both the standard and the gap splits [20] on the HOUSING and ENERGY datasets from the UCI machine learning repository [16]. On ENERGY, we can reproduce the catastrophic failure of VI both with BBB and Rank-1 VI, but not with iVON which performs still similarly to MultiSWAG. Overall, we find that the benefit of ensembles is less clear than on the larger WILDS datasets, which emphasizes the importance of evaluating Bayesian algorithms on large datasets.

**Hyperparameters.** All hyperparameters were optimized through a grid search on the validation set. Note that for the gap splits the validation set is not part of the gap. We considered 40, 100 and 200 epochs, learning rates of 0.01 and 0.001 and (where applicable) weight decay factors of $10^{-4}$ and $10^{-5}$. For BBB, the prior standard deviations are 0.1, 1.0 and 10.0 and we scale the KL divergence in the ELBO by 0.2, 0.5, and 1.0, with colder temperatures generally leading to better results. For iVON, we consider prior precisions of 10, 100, and 200, with 200 being selected in most cases. BBB and iVON use five Monte Carlo samples during training. For SWAG, we consider 60, 100, and 150 epochs, use 30 parameter samples and start sampling after 50%, 75%, or 90% of the training epochs were completed. For Laplace, we always use a last-layer approximation with a full covariance matrix. We use the Adam optimizer [43] to optimize the log-likelihood/ELBO and learn the output standard deviation jointly with the parameters. We use 1000 parameter samples for each prediction.

| Model | LML | MSE | QCE | sQCE |
|---|---|---|---|---|
| MAP | $-2.643 \pm 0.054$ | $10.707 \pm 0.794$ | $\mathbf{0.036 \pm 0.001}$ | $\mathbf{-0.018 \pm 0.012}$ |
| Deep Ensemble | $\mathbf{-2.330 \pm 0.015}$ | $6.034 \pm 0.413$ | $0.099 \pm 0.006$ | $0.099 \pm 0.006$ |
| MCD | $-2.836 \pm 0.095$ | $13.531 \pm 1.401$ | $0.044 \pm 0.008$ | $-0.031 \pm 0.019$ |
| MultiMCD | $\mathbf{-2.328 \pm 0.010}$ | $\mathbf{5.789 \pm 0.077}$ | $0.101 \pm 0.002$ | $0.101 \pm 0.002$ |
| SWAG | $-2.495 \pm 0.014$ | $6.044 \pm 0.214$ | $0.128 \pm 0.003$ | $0.128 \pm 0.003$ |
| MultiSWAG | $-2.511 \pm 0.002$ | $7.071 \pm 0.011$ | $0.139 \pm 0.002$ | $0.139 \pm 0.002$ |
| LL Laplace | $-2.515 \pm 0.058$ | $9.498 \pm 2.347$ | $0.099 \pm 0.020$ | $0.099 \pm 0.020$ |
| LL MultiLaplace | $-2.467 \pm 0.031$ | $\mathbf{5.850 \pm 0.395}$ | $0.157 \pm 0.005$ | $0.157 \pm 0.005$ |
| BBB | $-2.475 \pm 0.096$ | $7.716 \pm 1.053$ | $\mathbf{0.033 \pm 0.004}$ | $\mathbf{-0.006 \pm 0.012}$ |
| MultiBBB | $-2.529 \pm 0.003$ | $7.212 \pm 0.160$ | $0.172 \pm 0.005$ | $0.172 \pm 0.005$ |
| Rank-1 VI | $-2.531 \pm 0.103$ | $8.983 \pm 1.451$ | $\mathbf{0.033 \pm 0.008}$ | $\mathbf{0.006 \pm 0.016}$ |
| iVON | $-2.793 \pm 0.006$ | $9.853 \pm 0.318$ | $0.215 \pm 0.003$ | $0.215 \pm 0.003$ |
| SVGD | $-2.614 \pm 0.017$ | $8.397 \pm 0.642$ | $0.143 \pm 0.010$ | $0.143 \pm 0.010$ |

Table 4: UCI-HOUSING (standard splits)

| Model | LML | MSE | QCE | sQCE |
|---|---|---|---|---|
| MAP | $-2.850 \pm 0.207$ | $14.775 \pm 3.635$ | $\mathbf{0.040 \pm 0.013}$ | $\mathbf{-0.012 \pm 0.024}$ |
| Deep Ensemble | $-2.767 \pm 0.183$ | $\mathbf{13.012 \pm 3.034}$ | $0.054 \pm 0.011$ | $0.045 \pm 0.017$ |
| MCD | $-2.892 \pm 0.210$ | $15.488 \pm 3.661$ | $\mathbf{0.034 \pm 0.007}$ | $\mathbf{-0.001 \pm 0.018}$ |
| MultiMCD | $\mathbf{-2.730 \pm 0.132}$ | $\mathbf{12.760 \pm 2.838}$ | $0.054 \pm 0.013$ | $0.045 \pm 0.019$ |
| SWAG | $-2.743 \pm 0.042$ | $\mathbf{12.940 \pm 1.742}$ | $0.113 \pm 0.017$ | $0.112 \pm 0.017$ |
| MultiSWAG | $\mathbf{-2.694 \pm 0.047}$ | $\mathbf{11.941 \pm 1.874}$ | $0.118 \pm 0.020$ | $0.117 \pm 0.020$ |
| LL Laplace | $-2.873 \pm 0.117$ | $15.294 \pm 3.687$ | $0.074 \pm 0.018$ | $0.072 \pm 0.020$ |
| LL MultiLaplace | $-2.832 \pm 0.106$ | $\mathbf{13.445 \pm 3.209}$ | $0.106 \pm 0.020$ | $0.104 \pm 0.022$ |
| BBB | $-3.829 \pm 1.009$ | $17.238 \pm 7.435$ | $0.114 \pm 0.024$ | $-0.113 \pm 0.024$ |
| MultiBBB | $\mathbf{-2.734 \pm 0.115}$ | $14.071 \pm 2.925$ | $0.065 \pm 0.018$ | $0.054 \pm 0.026$ |
| Rank-1 VI | $-2.806 \pm 0.193$ | $\mathbf{13.513 \pm 2.930}$ | $0.067 \pm 0.025$ | $\mathbf{0.018 \pm 0.043}$ |
| iVON | $-2.930 \pm 0.023$ | $17.904 \pm 2.300$ | $0.152 \pm 0.015$ | $0.151 \pm 0.015$ |
| SVGD | $-2.855 \pm 0.184$ | $14.848 \pm 3.170$ | $\mathbf{0.039 \pm 0.014}$ | $\mathbf{-0.003 \pm 0.025}$ |

Table 5: UCI-HOUSING (gap splits)

| Model | LML | MSE | QCE | sQCE |
|---|---|---|---|---|
| MAP | $-1.702 \pm 0.094$ | $1.760 \pm 0.289$ | $\mathbf{0.051 \pm 0.020}$ | $\mathbf{-0.020 \pm 0.036}$ |
| Deep Ensemble | $-1.235 \pm 0.003$ | $\mathbf{0.177 \pm 0.007}$ | $0.270 \pm 0.002$ | $0.270 \pm 0.002$ |
| MCD | $-1.709 \pm 0.079$ | $1.779 \pm 0.252$ | $\mathbf{0.049 \pm 0.016}$ | $\mathbf{-0.022 \pm 0.032}$ |
| MultiMCD | $-1.236 \pm 0.005$ | $0.212 \pm 0.015$ | $0.260 \pm 0.003$ | $0.260 \pm 0.003$ |
| SWAG | $-2.127 \pm 0.029$ | $2.198 \pm 0.274$ | $0.210 \pm 0.006$ | $0.210 \pm 0.006$ |
| MultiSWAG | $-2.143 \pm 0.002$ | $2.454 \pm 0.018$ | $0.220 \pm 0.001$ | $0.220 \pm 0.001$ |
| LL Laplace | $-1.653 \pm 0.026$ | $0.608 \pm 0.110$ | $0.245 \pm 0.019$ | $0.245 \pm 0.019$ |
| LL MultiLaplace | $-1.606 \pm 0.016$ | $0.235 \pm 0.033$ | $0.316 \pm 0.008$ | $0.316 \pm 0.008$ |
| BBB | $\mathbf{-0.976 \pm 0.123}$ | $0.413 \pm 0.103$ | $\mathbf{0.055 \pm 0.017}$ | $\mathbf{0.030 \pm 0.032}$ |
| MultiBBB | $\mathbf{-1.022 \pm 0.021}$ | $0.309 \pm 0.075$ | $0.210 \pm 0.012$ | $0.210 \pm 0.012$ |
| Rank-1 VI | $\mathbf{-1.029 \pm 0.166}$ | $0.459 \pm 0.145$ | $\mathbf{0.054 \pm 0.019}$ | $\mathbf{0.019 \pm 0.036}$ |
| iVON | $-2.463 \pm 0.006$ | $6.620 \pm 0.191$ | $0.161 \pm 0.010$ | $0.161 \pm 0.010$ |
| SVGD | $-1.322 \pm 0.040$ | $0.550 \pm 0.121$ | $0.159 \pm 0.027$ | $0.159 \pm 0.027$ |

Table 6: UCI-ENERGY (standard splits)

| Model | LML | MSE | QCE | sQCE |
|---|---|---|---|---|
| MAP | $-7.723 \pm 7.553$ | $\mathbf{34.444 \pm 41.620}$ | $0.247 \pm 0.065$ | $\mathbf{0.043 \pm 0.195}$ |
| Deep Ensemble | $-4.360 \pm 3.066$ | $\mathbf{31.419 \pm 36.845}$ | $0.272 \pm 0.060$ | $\mathbf{0.072 \pm 0.207}$ |
| MCD | $-10.299 \pm 10.685$ | $48.491 \pm 58.682$ | $0.261 \pm 0.065$ | $\mathbf{0.041 \pm 0.206}$ |
| MultiMCD | $-6.744 \pm 6.151$ | $41.030 \pm 49.269$ | $0.272 \pm 0.061$ | $\mathbf{0.073 \pm 0.207}$ |
| SWAG | $\mathbf{-3.655 \pm 1.469}$ | $\mathbf{30.372 \pm 28.902}$ | $0.218 \pm 0.084$ | $\mathbf{0.011 \pm 0.183}$ |
| MultiSWAG | $\mathbf{-3.110 \pm 0.815}$ | $\mathbf{25.362 \pm 22.428}$ | $0.192 \pm 0.059$ | $\mathbf{0.034 \pm 0.152}$ |
| LL Laplace | $-7.009 \pm 4.256$ | $45.505 \pm 37.787$ | $0.247 \pm 0.040$ | $\mathbf{0.116 \pm 0.119}$ |
| LL MultiLaplace | $-5.549 \pm 2.983$ | $38.452 \pm 31.657$ | $0.270 \pm 0.046$ | $\mathbf{0.142 \pm 0.127}$ |
| BBB | $-64.268 \pm 79.182$ | $43.670 \pm 52.833$ | $0.199 \pm 0.131$ | $\mathbf{-0.101 \pm 0.184}$ |
| MultiBBB | $-22.150 \pm 25.068$ | $50.502 \pm 58.432$ | $0.236 \pm 0.110$ | $\mathbf{-0.073 \pm 0.202}$ |
| Rank-1 VI | $-72.412 \pm 92.191$ | $49.099 \pm 60.606$ | $0.191 \pm 0.133$ | $\mathbf{-0.109 \pm 0.178}$ |
| iVON | $\mathbf{-3.367 \pm 0.903}$ | $\mathbf{21.546 \pm 13.347}$ | $\mathbf{0.109 \pm 0.025}$ | $\mathbf{0.038 \pm 0.074}$ |
| SVGD | $-9.945 \pm 10.449$ | $46.757 \pm 56.551$ | $0.227 \pm 0.067$ | $\mathbf{0.037 \pm 0.182}$ |

Table 7: UCI-ENERGY (gap splits)

## G.2 CIFAR-10

Following Wilson et al. [90], we train a ResNet-20 [32] with Swish activations [73] and Filter Response Normalization [82]. The use of Filter Response Normalization instead of batch normalization, which only uses batch statistics, eliminates the problems mentioned in Appendix E. We train all models except iVON with SGD and a learning rate of $0.05$ and Nesterov momentum of strength $0.9$ for 300 epochs. We use the learning rate schedule from Maddox et al. [59]: The learning rate is kept at its initial value for the first 150 epochs, then linearly reduced to a learning rate of $0.005$ at epoch 270 at which it is kept constant for the remaining 30 epochs. For MCD, we use a dropout rate of $0.1$ and insert dropout units after every linear and convolutional layer of the ResNet-20. For BBB, we temper the KL divergence in the ELBO with a factor of $0.2$. Rank-1 VI uses an untempered posterior and four components. BBB and iVON use two Monte Carlo samples during training. The Laplace approximation is based on a diagonal last-layer approximation. iVON is also trained for 300 epochs with a learning rate of $1 \cdot 10^{-4}$, a prior precision of $50$, and a data augmentation factor of $10$ (see Osawa et al. [70] for details), but uses no learning rate schedule. We found these changes to be necessary to ensure that iVON performs well, likely because iVON is much more similar to Adam [43] than to SGD and therefore needs a smaller learning rate. Following Nado et al. [65], SNGP uses a spectral normalization factor of $6.0$ and mean field factor of $20$. We did not perform any additional tuning of the mean field factor. We always use 50 parameter samples during evaluation.

Figure 9 displays the accuracy, ECE, sECE, agreement with HMC, and TV compared to HMC. MultiX models tend to become underconfident. Table 8 shows detailed numerical results for all algorithms and corruption levels.

| Model | Accuracy | ECE | sECE | NLL | Agreement | TV |
|---|---|---|---|---|---|---|
| MAP | $0.925 \pm 0.001$ | $0.045 \pm 0.001$ | $-0.045 \pm 0.001$ | $0.296 \pm 0.006$ | $0.906 \pm 0.002$ | $0.172 \pm 0.001$ |
| Deep Ensemble | $\mathbf{0.944 \pm 0.001}$ | $0.010 \pm 0.001$ | $\mathbf{0.003 \pm 0.000}$ | $\mathbf{0.174 \pm 0.001}$ | $0.923 \pm 0.001$ | $0.144 \pm 0.001$ |
| MCD | $0.927 \pm 0.002$ | $\mathbf{0.008 \pm 0.001}$ | $0.007 \pm 0.001$ | $0.216 \pm 0.005$ | $0.920 \pm 0.003$ | $0.132 \pm 0.002$ |
| MultiMCD | $0.941 \pm 0.001$ | $0.031 \pm 0.001$ | $0.031 \pm 0.001$ | $0.186 \pm 0.001$ | $0.930 \pm 0.002$ | $0.116 \pm 0.001$ |
| SWAG | $0.921 \pm 0.002$ | $0.042 \pm 0.003$ | $0.042 \pm 0.003$ | $0.250 \pm 0.002$ | $0.910 \pm 0.002$ | $0.130 \pm 0.001$ |
| MultiSWAG | $0.940 \pm 0.001$ | $0.099 \pm 0.002$ | $0.099 \pm 0.002$ | $0.258 \pm 0.002$ | $0.927 \pm 0.001$ | $0.107 \pm 0.001$ |
| LL Laplace | $0.924 \pm 0.001$ | $0.040 \pm 0.001$ | $-0.040 \pm 0.001$ | $0.282 \pm 0.006$ | $0.906 \pm 0.002$ | $0.169 \pm 0.001$ |
| LL MultiLaplace | $\mathbf{0.945 \pm 0.001}$ | $0.012 \pm 0.001$ | $0.007 \pm 0.001$ | $\mathbf{0.174 \pm 0.002}$ | $0.923 \pm 0.001$ | $0.142 \pm 0.000$ |
| BBB | $0.898 \pm 0.003$ | $0.046 \pm 0.002$ | $-0.046 \pm 0.002$ | $0.387 \pm 0.011$ | $0.900 \pm 0.003$ | $0.161 \pm 0.001$ |
| MultiBBB | $0.929 \pm 0.001$ | $0.018 \pm 0.002$ | $0.018 \pm 0.002$ | $0.228 \pm 0.002$ | $0.930 \pm 0.001$ | $0.122 \pm 0.001$ |
| Rank1-VI | $0.881 \pm 0.003$ | $0.041 \pm 0.002$ | $0.041 \pm 0.002$ | $0.363 \pm 0.005$ | $0.910 \pm 0.003$ | $0.116 \pm 0.002$ |
| iVON | $0.842 \pm 0.004$ | $0.025 \pm 0.003$ | $0.024 \pm 0.003$ | $0.464 \pm 0.011$ | $0.874 \pm 0.003$ | $0.145 \pm 0.003$ |
| MultiiVON | $0.881 \pm 0.003$ | $0.077 \pm 0.003$ | $0.077 \pm 0.003$ | $0.388 \pm 0.002$ | $0.921 \pm 0.002$ | $0.107 \pm 0.001$ |
| SVGD | $0.927 \pm 0.001$ | $0.018 \pm 0.001$ | $\mathbf{0.003 \pm 0.001}$ | $0.255 \pm 0.001$ | $0.924 \pm 0.002$ | $0.135 \pm 0.001$ |
| SNGP | $0.917 \pm 0.003$ | $0.076 \pm 0.006$ | $0.076 \pm 0.006$ | $0.380 \pm 0.013$ | $0.903 \pm 0.002$ | $0.178 \pm 0.002$ |
| HMC | $0.903$ | $0.069$ | $0.068$ | $0.320$ | $1.000$ | $0.000$ |

(a) Standard Evaluation Split (Corruption Level 0)

| Model | Accuracy | ECE | sECE | NLL | Agreement | TV |
|---|---|---|---|---|---|---|
| MAP | $0.872 \pm 0.002$ | $0.080 \pm 0.002$ | $-0.080 \pm 0.002$ | $0.518 \pm 0.010$ | $0.848 \pm 0.001$ | $0.238 \pm 0.001$ |
| Deep Ensemble | $\mathbf{0.903 \pm 0.001}$ | $0.014 \pm 0.001$ | $-0.003 \pm 0.001$ | $0.305 \pm 0.001$ | $0.870 \pm 0.003$ | $0.196 \pm 0.001$ |
| MCD | $0.872 \pm 0.004$ | $\mathbf{0.011 \pm 0.003}$ | $\mathbf{-0.004 \pm 0.006}$ | $0.400 \pm 0.011$ | $0.865 \pm 0.003$ | $0.184 \pm 0.002$ |
| MultiMCD | $0.890 \pm 0.003$ | $0.028 \pm 0.002$ | $0.026 \pm 0.002$ | $0.335 \pm 0.005$ | $0.880 \pm 0.002$ | $0.158 \pm 0.001$ |
| SWAG | $0.876 \pm 0.004$ | $0.047 \pm 0.006$ | $0.047 \pm 0.005$ | $0.388 \pm 0.004$ | $0.856 \pm 0.005$ | $0.179 \pm 0.001$ |
| MultiSWAG | $0.900 \pm 0.001$ | $0.117 \pm 0.002$ | $0.117 \pm 0.002$ | $0.383 \pm 0.002$ | $0.878 \pm 0.001$ | $0.147 \pm 0.000$ |
| LL Laplace | $0.873 \pm 0.004$ | $0.074 \pm 0.004$ | $-0.074 \pm 0.004$ | $0.499 \pm 0.017$ | $0.849 \pm 0.002$ | $0.239 \pm 0.002$ |
| LL MultiLaplace | $\mathbf{0.903 \pm 0.005}$ | $0.016 \pm 0.003$ | $\mathbf{-0.001 \pm 0.005}$ | $0.314 \pm 0.003$ | $0.859 \pm 0.004$ | $0.199 \pm 0.001$ |
| BBB | $0.839 \pm 0.003$ | $0.083 \pm 0.003$ | $-0.083 \pm 0.003$ | $0.682 \pm 0.029$ | $0.844 \pm 0.002$ | $0.222 \pm 0.000$ |
| MultiBBB | $0.878 \pm 0.008$ | $0.018 \pm 0.005$ | $0.006 \pm 0.009$ | $0.394 \pm 0.012$ | $0.880 \pm 0.005$ | $0.170 \pm 0.004$ |
| Rank1-VI | $0.843 \pm 0.004$ | $0.042 \pm 0.004$ | $0.042 \pm 0.004$ | $0.484 \pm 0.015$ | $0.860 \pm 0.002$ | $0.165 \pm 0.003$ |
| iVON | $0.795 \pm 0.010$ | $0.025 \pm 0.006$ | $0.010 \pm 0.010$ | $0.596 \pm 0.021$ | $0.827 \pm 0.011$ | $0.189 \pm 0.008$ |
| MultiiVON | $0.845 \pm 0.004$ | $0.083 \pm 0.003$ | $0.081 \pm 0.003$ | $0.504 \pm 0.018$ | $0.863 \pm 0.008$ | $0.145 \pm 0.003$ |
| SVGD | $0.883 \pm 0.001$ | $0.021 \pm 0.004$ | $-0.007 \pm 0.001$ | $0.432 \pm 0.010$ | $0.875 \pm 0.001$ | $0.192 \pm 0.000$ |
| SNGP | $0.867 \pm 0.015$ | $0.074 \pm 0.009$ | $0.069 \pm 0.012$ | $0.560 \pm 0.034$ | $0.844 \pm 0.010$ | $0.231 \pm 0.004$ |
| HMC | $0.834$ | $0.066$ | $0.064$ | $0.508$ | $1.000$ | $0.000$ |

(b) Corruption Level 1

| Model | Accuracy | ECE | sECE | NLL | Agreement | TV |
|---|---|---|---|---|---|---|
| MAP | $0.805 \pm 0.005$ | $0.128 \pm 0.005$ | $-0.128 \pm 0.004$ | $0.863 \pm 0.019$ | $0.778 \pm 0.002$ | $0.309 \pm 0.002$ |
| Deep Ensemble | $\mathbf{0.838 \pm 0.004}$ | $0.027 \pm 0.004$ | $-0.027 \pm 0.004$ | $\mathbf{0.518 \pm 0.020}$ | $0.805 \pm 0.002$ | $0.253 \pm 0.001$ |
| MCD | $0.777 \pm 0.011$ | $0.047 \pm 0.004$ | $-0.047 \pm 0.005$ | $0.733 \pm 0.051$ | $0.796 \pm 0.010$ | $0.232 \pm 0.001$ |
| MultiMCD | $0.805 \pm 0.003$ | $\mathbf{0.011 \pm 0.003}$ | $\mathbf{0.000 \pm 0.003}$ | $0.594 \pm 0.012$ | $0.823 \pm 0.005$ | $0.196 \pm 0.001$ |
| SWAG | $0.806 \pm 0.004$ | $0.032 \pm 0.003$ | $0.031 \pm 0.004$ | $0.593 \pm 0.009$ | $0.783 \pm 0.005$ | $0.228 \pm 0.003$ |
| MultiSWAG | $\mathbf{0.839 \pm 0.002}$ | $0.117 \pm 0.004$ | $0.117 \pm 0.004$ | $0.556 \pm 0.002$ | $0.818 \pm 0.001$ | $0.186 \pm 0.001$ |
| LL Laplace | $0.804 \pm 0.003$ | $0.120 \pm 0.002$ | $-0.119 \pm 0.002$ | $0.839 \pm 0.021$ | $0.777 \pm 0.005$ | $0.308 \pm 0.003$ |
| LL MultiLaplace | $\mathbf{0.850 \pm 0.012}$ | $0.026 \pm 0.008$ | $-0.015 \pm 0.009$ | $\mathbf{0.498 \pm 0.035}$ | $0.800 \pm 0.002$ | $0.251 \pm 0.003$ |
| BBB | $0.735 \pm 0.009$ | $0.154 \pm 0.008$ | $-0.154 \pm 0.008$ | $1.296 \pm 0.072$ | $0.774 \pm 0.004$ | $0.286 \pm 0.002$ |
| MultiBBB | $0.786 \pm 0.014$ | $0.033 \pm 0.010$ | $-0.026 \pm 0.014$ | $0.741 \pm 0.045$ | $0.830 \pm 0.006$ | $0.204 \pm 0.003$ |
| Rank1-VI | $0.774 \pm 0.008$ | $0.024 \pm 0.002$ | $0.019 \pm 0.006$ | $0.684 \pm 0.027$ | $0.802 \pm 0.006$ | $0.210 \pm 0.005$ |
| iVON | $0.725 \pm 0.014$ | $0.028 \pm 0.004$ | $-0.022 \pm 0.005$ | $0.809 \pm 0.036$ | $0.756 \pm 0.016$ | $0.237 \pm 0.008$ |
| MultiiVON | $0.783 \pm 0.008$ | $0.069 \pm 0.009$ | $0.068 \pm 0.010$ | $0.666 \pm 0.006$ | $0.821 \pm 0.002$ | $0.179 \pm 0.003$ |
| SVGD | $0.804 \pm 0.004$ | $0.038 \pm 0.002$ | $-0.038 \pm 0.002$ | $0.821 \pm 0.024$ | $0.818 \pm 0.001$ | $0.238 \pm 0.001$ |
| SNGP | $0.785 \pm 0.015$ | $0.067 \pm 0.007$ | $0.044 \pm 0.010$ | $0.837 \pm 0.042$ | $0.767 \pm 0.008$ | $0.295 \pm 0.004$ |
| HMC | $0.724$ | $0.020$ | $0.017$ | $0.833$ | $1.000$ | $0.000$ |

(c) Corruption Level 3

| Model | Accuracy | ECE | sECE | NLL | Agreement | TV |
|---|---|---|---|---|---|---|
| MAP | $0.689 \pm 0.006$ | $0.217 \pm 0.006$ | $-0.217 \pm 0.006$ | $1.494 \pm 0.019$ | $0.683 \pm 0.005$ | $0.390 \pm 0.003$ |
| Deep Ensemble | $\mathbf{0.733 \pm 0.006}$ | $0.075 \pm 0.007$ | $-0.075 \pm 0.007$ | $0.937 \pm 0.036$ | $0.718 \pm 0.004$ | $0.310 \pm 0.003$ |
| MCD | $0.629 \pm 0.009$ | $0.141 \pm 0.011$ | $-0.141 \pm 0.010$ | $1.312 \pm 0.094$ | $0.725 \pm 0.012$ | $0.277 \pm 0.003$ |
| MultiMCD | $0.666 \pm 0.007$ | $0.069 \pm 0.007$ | $-0.069 \pm 0.007$ | $1.063 \pm 0.021$ | $0.760 \pm 0.001$ | $0.234 \pm 0.001$ |
| SWAG | $0.696 \pm 0.005$ | $\mathbf{0.018 \pm 0.009}$ | $-0.012 \pm 0.008$ | $0.927 \pm 0.025$ | $0.699 \pm 0.006$ | $0.277 \pm 0.006$ |
| MultiSWAG | $\mathbf{0.728 \pm 0.004}$ | $0.082 \pm 0.004$ | $0.082 \pm 0.004$ | $\mathbf{0.841 \pm 0.009}$ | $0.740 \pm 0.004$ | $0.228 \pm 0.002$ |
| LL Laplace | $0.690 \pm 0.006$ | $0.203 \pm 0.005$ | $-0.203 \pm 0.005$ | $1.430 \pm 0.026$ | $0.685 \pm 0.003$ | $0.380 \pm 0.002$ |
| LL MultiLaplace | $\mathbf{0.731 \pm 0.013}$ | $0.073 \pm 0.011$ | $-0.072 \pm 0.011$ | $0.941 \pm 0.064$ | $0.722 \pm 0.008$ | $0.303 \pm 0.002$ |
| BBB | $0.584 \pm 0.011$ | $0.268 \pm 0.014$ | $-0.268 \pm 0.014$ | $2.413 \pm 0.126$ | $0.698 \pm 0.008$ | $0.353 \pm 0.003$ |
| MultiBBB | $0.621 \pm 0.014$ | $0.114 \pm 0.016$ | $-0.114 \pm 0.016$ | $1.411 \pm 0.105$ | $0.757 \pm 0.010$ | $0.247 \pm 0.009$ |
| Rank1-VI | $0.673 \pm 0.016$ | $0.028 \pm 0.010$ | $-0.024 \pm 0.014$ | $1.010 \pm 0.056$ | $0.719 \pm 0.013$ | $0.262 \pm 0.010$ |
| iVON | $0.617 \pm 0.011$ | $0.086 \pm 0.006$ | $-0.086 \pm 0.006$ | $1.201 \pm 0.061$ | $0.678 \pm 0.008$ | $0.290 \pm 0.007$ |
| MultiiVON | $0.657 \pm 0.012$ | $\mathbf{0.025 \pm 0.005}$ | $\mathbf{0.000 \pm 0.009}$ | $0.998 \pm 0.047$ | $0.752 \pm 0.012$ | $0.224 \pm 0.006$ |
| SVGD | $0.666 \pm 0.004$ | $0.117 \pm 0.009$ | $-0.117 \pm 0.009$ | $1.557 \pm 0.043$ | $0.742 \pm 0.007$ | $0.286 \pm 0.002$ |
| SNGP | $0.657 \pm 0.006$ | $0.084 \pm 0.007$ | $-0.017 \pm 0.008$ | $1.256 \pm 0.022$ | $0.681 \pm 0.008$ | $0.353 \pm 0.005$ |
| HMC | $0.592$ | $0.055$ | $-0.054$ | $1.225$ | $1.000$ | $0.000$ |

(d) Corruption Level 5

Table 8: CIFAR-10: Detailed results on the standard evaluation split and the corruption levels 1, 3, and 5 of CIFAR-10-C.

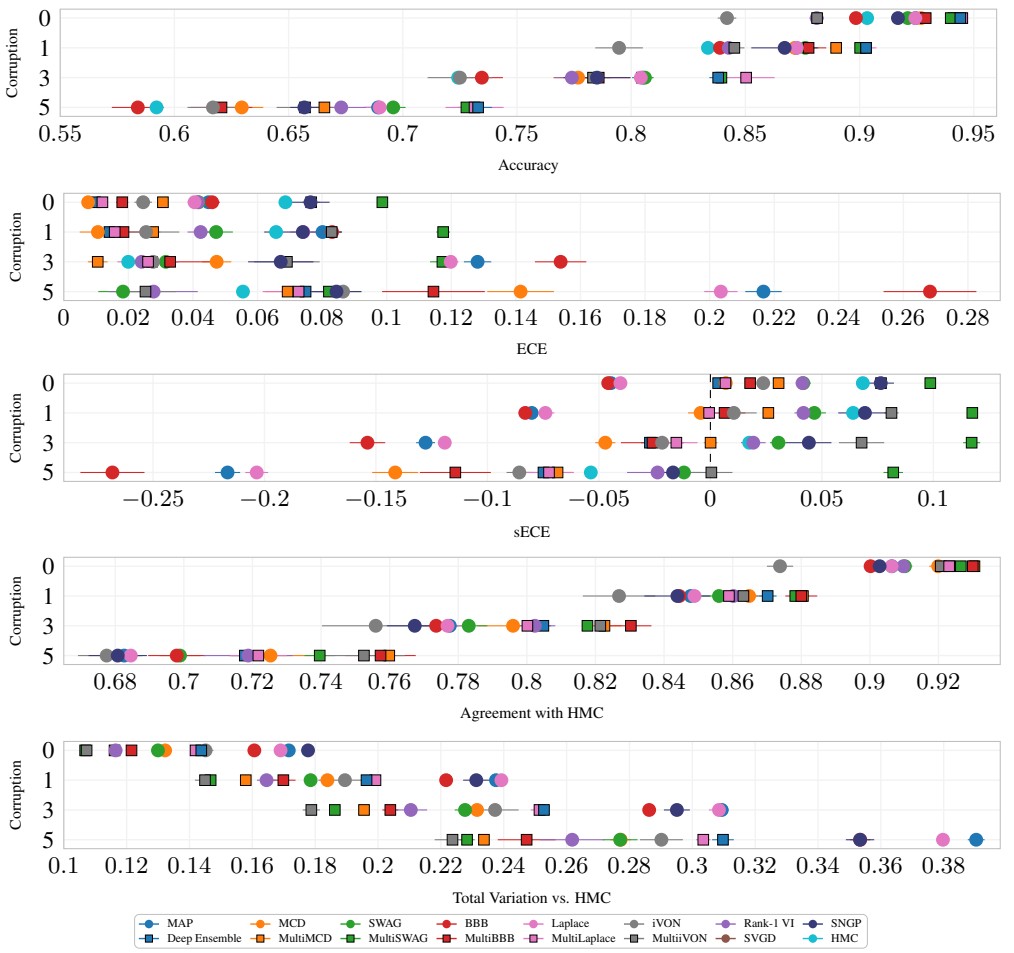

Figure 9: CIFAR-10-(C): All results for the corruption intensities 0, 1, 3, and 5. The corruption intensities are denoted on the y-axis. A negative sECE indicates overconfidence, a positive sECE indicates underconfidence. The plot for the TV is repeated from Figure 5.

## G.3 WILDS

We strictly follow the training and evaluation protocol of Koh et al. [47] by reusing their data folds for training, validation, and testing. We use the hyperparameters proposed by Koh et al. [47] where applicable, and set the other hyperparameters to standard values as suggested by the developers of the respective algorithms. If the standard values lead to unexpectedly bad results, we tune the hyperparameters through a grid search. Hyperparameter tuning was performed on the i.d. validation and, where available, o.o.d. validation splits, but never on testing splits. In particular, we select the prior precision of iVON through a grid search over the values $1, 10, 100,$ and $500$ per model architecture. We find the prior precision of iVON to be hard to tune, as iVON frequently diverges for comparatively small prior precisions such as $1$ and $10$. BBB works always well with the standard unit prior. We also experiment with other priors but find no difference in performance except on RxRx1-WILDS (see Figure 14). BBB and iVON use two Monte Carlo samples during training. See the sections below for the hyperparameters that were chosen on the individual datasets. We use mixed precision training whenever possible. The VI algorithms as well as the Laplace approximations are mostly trained without mixed precision, as this leads to unstable training.

SNGP uses the same learning rate, weight decay, and number of epochs as the other algorithms. Following the recommendations by Liu et al. [54] and the tuning done by Nado et al. [65], we use a spectral normalization factor of $6.0$ for the computer vision tasks and $0.95$ for the text classification tasks. On the image classification tasks, SNGP performs significantly better when limiting the input

dimension of the Gaussian Process to 128 or 256 instead of using the output dimension of the previous network layer.

We use 10 posterior samples per prediction during evaluation to constrain the computational overhead of the Bayesian algorithms, which is generally sufficient to capture the predictive distribution [72]. Note that our results are not directly comparable to the results of Daxberger et al. [13], as they build their Deep Ensembles and Laplace approximations from the pretrained models provided by Koh et al. [47]. When comparing our results with the best performing algorithms on the WILDS leaderboard, we only consider the algorithms on the "overall leaderboard", i.e. the algorithms that conform to the official submission guidelines of Koh et al. [47].

### G.3.1 Camelyon17-WILDS

Following Koh et al. [47], we train a DenseNet-121 [35] with SGD for 5 epochs with a learning rate of 0.001, weight decay 0.01 and momentum 0.9. SWAG collects 30 parameter samples during the last epoch.

### G.3.2 PovertyMAP-WILDS

We train a ResNet-18 [32] using the same hyperparameters as Koh et al. [47] where applicable: A learning rate of $10^{-3}$ and no weight decay. We only train for 100 epochs as all models were converged after that. SWAG collects 30 parameter samples starting at epoch 50. For BBB, we scale the KL divergence down with a factor of 0.2, as this significantly improves the MSE. Rank-1 VI uses an unscaled KL divergence. The ensembles, Rank-1 VI and SVGD use five members/components. We optimize the log likelihood of the training data and represent the aleatoric uncertainty with a fixed standard deviation of 0.1, as this is the value MAP converges to when jointly optimizing the standard deviation and the model's parameters. For the final evaluations, we do not optimize the standard deviation, as this leads to unstable training with the VI algorithms. Following Koh et al. [47], we aggregate all results over the five folds of POVERTYMAP, with one seed per fold.

As mentioned in the main paper, iVON performs significantly worse than the other algorithms. We conducted a grid search over prior precisions 1, 10, 100 and 500 with a single seed per value, and found that for 1 and 10 iVON diverges, for 100 iVON achieves an o.o.d. Pearson coefficient on the "A" split of 0.21 and for 500 it achieves a Pearson coefficient of 0.25. Most likely due to their underfitting the non-diverged models are comparatively well calibrated with sECEs of $-0.21$ for a prior precision of 100 and $-0.24$ for a prior precision of 500.

**Log Marginal Likelihood.** The log marginal likelihood is commonly used to jointly evaluate the accuracy and calibration of a regression model. On an evaluation dataset $\mathcal{D}'$, the log marginal likelihood (LML) is given by

$$\text{LML} = \log p(\mathcal{D}' \mid \mathcal{D}) = \log \int p(\mathcal{D}' \mid \boldsymbol{\theta})p(\boldsymbol{\theta} \mid \mathcal{D}) \, \mathrm{d}\boldsymbol{\theta} \approx \log \sum_n p(\mathcal{D}' \mid \boldsymbol{\theta}_n), \quad (14)$$

where the $\theta_n$ are samples from the parameter posterior. When only few predictions are available because sampling parameters $\theta_n$ or evaluating the likelihood $p(\mathcal{D}' \mid \boldsymbol{\theta}_n)$ is expensive, the LML may become very noisy. We therefore also report the per-sample log marginal likelihood

$$
\begin{aligned}
\text{psLML} &= \sum_{(\boldsymbol{x}_i, \boldsymbol{y}_i) \in \mathcal{D}'} \log p(\boldsymbol{y}_i \mid \boldsymbol{x}_i, \mathcal{D}) \\
&= \sum_{(\boldsymbol{x}_i, \boldsymbol{y}_i) \in \mathcal{D}'} \log \int p(\boldsymbol{y}_i \mid \boldsymbol{x}_i, \boldsymbol{\theta})p(\boldsymbol{\theta} \mid \mathcal{D}) \, \mathrm{d}\boldsymbol{\theta} \\
&\approx \sum_{(\boldsymbol{x}_i, \boldsymbol{y}_i) \in \mathcal{D}'} \log \sum_n p(\boldsymbol{y}_i \mid \boldsymbol{x}_i, \boldsymbol{\theta}_n),
\end{aligned}
\quad (15)
$$

which has a lower variance than the LML. We present the results for the LML, the psLML, the urban/rural Pearson coefficient (see Section 5.1), and the sQCE in Figure 10 and Figure 11. Table 9 shows detailed numerical results.

### (a) O.o.d. Evaluation Split

| Model | Worst U/R Pearson | psLML | LML | MSE | QCE | sQCE |
|---|---|---|---|---|---|---|
| MAP | $\mathbf{0.487 \pm 0.074}$ | $-11.945 \pm 2.042$ | $\mathbf{-11.945 \pm 2.042}$ | $\mathbf{0.267 \pm 0.041}$ | $0.382 \pm 0.012$ | $-0.382 \pm 0.012$ |
| Deep Ensemble | $\mathbf{0.520 \pm 0.075}$ | $-6.126 \pm 1.422$ | $\mathbf{-12.113 \pm 2.074}$ | $\mathbf{0.249 \pm 0.043}$ | $0.283 \pm 0.027$ | $-0.283 \pm 0.027$ |
| MCD | $\mathbf{0.491 \pm 0.079}$ | $-6.868 \pm 1.720$ | $\mathbf{-12.175 \pm 2.398}$ | $\mathbf{0.259 \pm 0.049}$ | $0.316 \pm 0.023$ | $-0.316 \pm 0.023$ |
| MultiMCD | $\mathbf{0.516 \pm 0.078}$ | $-5.053 \pm 1.518$ | $\mathbf{-12.290 \pm 2.409}$ | $\mathbf{0.253 \pm 0.052}$ | $0.271 \pm 0.035$ | $-0.271 \pm 0.035$ |
| SWAG | $\mathbf{0.473 \pm 0.078}$ | $-12.551 \pm 1.994$ | $\mathbf{-12.621 \pm 2.000}$ | $\mathbf{0.280 \pm 0.040}$ | $0.386 \pm 0.012$ | $-0.386 \pm 0.012$ |
| MultiSWAG | $\mathbf{0.512 \pm 0.078}$ | $-6.010 \pm 1.373$ | $\mathbf{-12.167 \pm 1.989}$ | $\mathbf{0.250 \pm 0.042}$ | $0.280 \pm 0.026$ | $-0.280 \pm 0.026$ |
| LL Laplace | $\mathbf{0.473 \pm 0.072}$ | $-12.599 \pm 2.191$ | $\mathbf{-12.599 \pm 2.191}$ | $\mathbf{0.280 \pm 0.044}$ | $0.387 \pm 0.012$ | $-0.387 \pm 0.012$ |
| LL MultiLaplace | $\mathbf{0.516 \pm 0.080}$ | $-5.614 \pm 1.271$ | $\mathbf{-12.324 \pm 2.141}$ | $\mathbf{0.251 \pm 0.044}$ | $0.265 \pm 0.026$ | $-0.265 \pm 0.026$ |
| BBB | $\mathbf{0.500 \pm 0.072}$ | $-7.881 \pm 2.290$ | $\mathbf{-12.075 \pm 2.470}$ | $\mathbf{0.264 \pm 0.054}$ | $0.333 \pm 0.036$ | $-0.333 \pm 0.036$ |
| MultiBBB | $\mathbf{0.518 \pm 0.074}$ | $-6.257 \pm 1.462$ | $\mathbf{-11.498 \pm 2.299}$ | $\mathbf{0.252 \pm 0.048}$ | $0.309 \pm 0.027$ | $-0.309 \pm 0.027$ |
| Rank-1 VI | $\mathbf{0.509 \pm 0.069}$ | $\mathbf{-3.568 \pm 1.053}$ | $\mathbf{-13.276 \pm 1.978}$ | $\mathbf{0.246 \pm 0.043}$ | $\mathbf{0.212 \pm 0.028}$ | $\mathbf{-0.212 \pm 0.028}$ |
| iVON | $0.249 \pm -$ | $-4.657 \pm -$ | $-19.787 \pm -$ | $0.347 \pm -$ | $0.236 \pm -$ | $-0.236 \pm -$ |
| SVGD | $\mathbf{0.497 \pm 0.070}$ | $-5.416 \pm 1.270$ | $\mathbf{-12.524 \pm 2.224}$ | $\mathbf{0.254 \pm 0.041}$ | $0.255 \pm 0.026$ | $-0.255 \pm 0.026$ |
| SNGP | $\mathbf{0.456 \pm 0.072}$ | $-12.688 \pm 1.556$ | $\mathbf{-12.688 \pm 1.556}$ | $\mathbf{0.281 \pm 0.031}$ | $0.357 \pm 0.009$ | $-0.357 \pm 0.009$ |

(a) O.o.d. Evaluation Split

### (b) I.d. Evaluation Split

| Model | Worst U/R Pearson | psLML | LML | MSE | QCE | sQCE |
|---|---|---|---|---|---|---|
| MAP | $0.673 \pm 0.019$ | $-7.445 \pm 0.761$ | $\mathbf{-7.445 \pm 0.761}$ | $0.177 \pm 0.015$ | $0.348 \pm 0.008$ | $-0.348 \pm 0.008$ |
| Deep Ensemble | $\mathbf{0.703 \pm 0.022}$ | $-3.438 \pm 0.569$ | $\mathbf{-7.093 \pm 0.744}$ | $\mathbf{0.155 \pm 0.014}$ | $0.228 \pm 0.010$ | $-0.228 \pm 0.010$ |
| MCD | $\mathbf{0.695 \pm 0.010}$ | $-3.604 \pm 0.483$ | $\mathbf{-7.237 \pm 0.673}$ | $\mathbf{0.162 \pm 0.012}$ | $0.267 \pm 0.014$ | $-0.267 \pm 0.014$ |
| MultiMCD | $\mathbf{0.711 \pm 0.024}$ | $-2.680 \pm 0.358$ | $\mathbf{-7.383 \pm 0.615}$ | $\mathbf{0.156 \pm 0.012}$ | $0.220 \pm 0.014$ | $-0.220 \pm 0.014$ |
| SWAG | $0.664 \pm 0.021$ | $-7.719 \pm 0.716$ | $\mathbf{-7.752 \pm 0.723}$ | $0.183 \pm 0.014$ | $0.355 \pm 0.003$ | $-0.355 \pm 0.003$ |
| MultiSWAG | $\mathbf{0.705 \pm 0.023}$ | $-3.331 \pm 0.439$ | $\mathbf{-7.221 \pm 0.625}$ | $\mathbf{0.155 \pm 0.012}$ | $0.223 \pm 0.008$ | $-0.223 \pm 0.008$ |
| LL Laplace | $0.664 \pm 0.018$ | $-7.823 \pm 0.737$ | $\mathbf{-7.823 \pm 0.737}$ | $0.184 \pm 0.015$ | $0.357 \pm 0.006$ | $-0.357 \pm 0.006$ |
| LL MultiLaplace | $\mathbf{0.702 \pm 0.023}$ | $-3.234 \pm 0.539$ | $\mathbf{-7.474 \pm 0.828}$ | $\mathbf{0.157 \pm 0.014}$ | $0.206 \pm 0.004$ | $-0.206 \pm 0.004$ |
| BBB | $0.680 \pm 0.019$ | $-4.508 \pm 0.269$ | $\mathbf{-7.224 \pm 0.754}$ | $0.169 \pm 0.014$ | $0.286 \pm 0.014$ | $-0.286 \pm 0.014$ |
| MultiBBB | $\mathbf{0.694 \pm 0.014}$ | $-3.619 \pm 0.344$ | $\mathbf{-7.159 \pm 0.586}$ | $\mathbf{0.161 \pm 0.011}$ | $0.256 \pm 0.012$ | $-0.256 \pm 0.012$ |
| Rank-1 VI | $0.669 \pm 0.011$ | $\mathbf{-2.077 \pm 0.323}$ | $-9.630 \pm 1.166$ | $0.173 \pm 0.016$ | $\mathbf{0.162 \pm 0.016}$ | $\mathbf{-0.162 \pm 0.016}$ |
| iVON | $0.571 \pm -$ | $-3.888 \pm -$ | $-15.832 \pm -$ | $0.284 \pm -$ | $0.201 \pm -$ | $-0.201 \pm -$ |
| SVGD | $\mathbf{0.694 \pm 0.026}$ | $-3.057 \pm 0.611$ | $\mathbf{-7.564 \pm 1.017}$ | $\mathbf{0.159 \pm 0.018}$ | $0.200 \pm 0.008$ | $-0.200 \pm 0.008$ |
| SNGP | $\mathbf{0.692 \pm 0.013}$ | $-7.135 \pm 0.639$ | $\mathbf{-7.135 \pm 0.639}$ | $0.170 \pm 0.013$ | $0.300 \pm 0.008$ | $-0.300 \pm 0.008$ |

(b) I.d. Evaluation Split

Table 9: POVERTYMAP-WILDS: Detailed results on the evaluation splits. iVON underperforms, with a Pearson coefficient of $0.249$ on the o.o.d. split and a Pearson coefficient of $0.571$ on the i.d. split. All models achieve the same LML and MSE within a $95\%$ confidence interval.

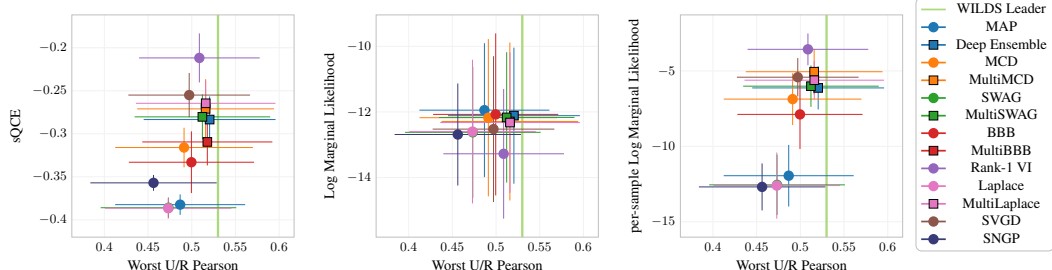

Figure 10: POVERTYMAP-WILDS: Worst urban/rural Pearson coefficient vs. sQCE, LML, and psLML on the o.o.d. evaluation split. Ensemble-based models consistently outperform single-mode models (note that Rank-1 VI's components and SVGD's particles give them ensemble-like properties). The psLML is less noisy than the LML and results in a ranking of the algorithms that is more consistent with the sQCE and the Pearson coefficient. Laplace and SWAG perform nearly equivalently, therefore the data points of SWAG are hidden behind the data points of Laplace. iVON performs significantly worse than the other algorithms and is therefore excluded.

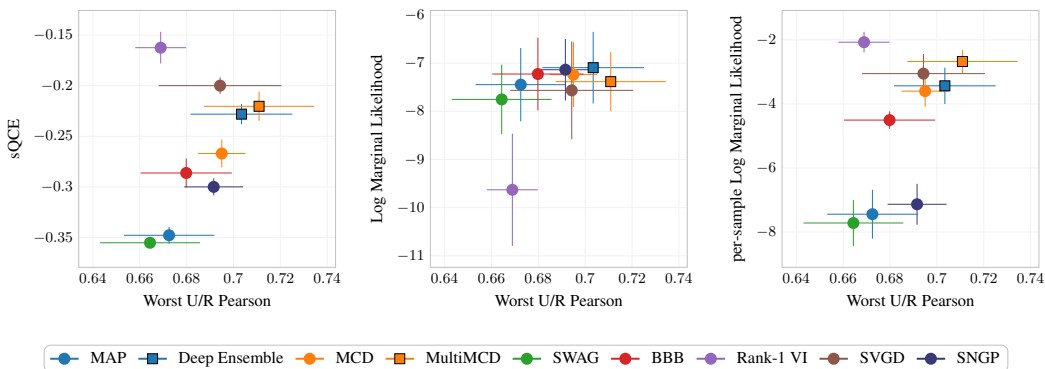

Figure 11: POVERTYMAP-WILDS: Worst urban/rural pearson coefficient vs. sQCE, LML and psLML on the i.d. evaluation split. The WILDS leaderboard [46] does not report the i.d. pearson coefficient.

### G.3.3 IWILDCAM-WILDS

Following Koh et al. [47], we finetune a ResNet-50 [32], pretrained on ImageNet [14], for 12 epochs with the Adam optimizer [43]. For each model, we replace the linear classification layer of the ResNet-50 by a randomly initialized one of the appropriate output dimension. We use the hyperparameters that Koh et al. [47] found to work best based on their grid search: A learning rate of $3 \cdot 10^{-5}$ and no weight decay. For MCD, we try dropout rates of $0.1$ and $0.2$ and select $0.1$ due to a slightly better macro F1 score on the evaluation split. iVON uses a prior precision of $100$, as optimized by a grid search. We use three seeds per model and build all ensembles by training six models independently and leaving out a different model for each of the three evaluation runs. Figure 12 shows the results on the o.o.d. evaluation split that are not presented in the main paper. Table 10 displays detailed numerical results on the o.o.d. evaluation split and on the i.d. validation split.

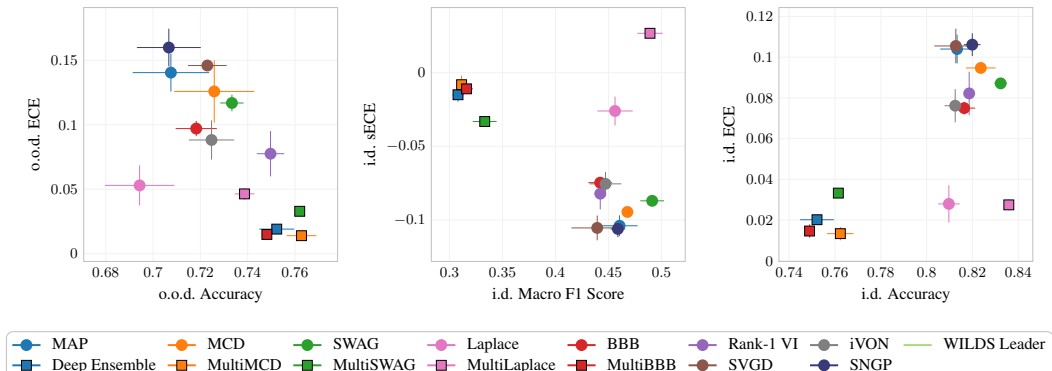

Figure 12: IWILDCAM-WILDS: Macro F1 score, accuracy, sECE and ECE on the o.o.d. evaluation split and the i.d. validation split (see Figure 2a for Macro F1 vs. sECE on the o.o.d. evaluation split). MultiX is less accurate than single-mode approximations on the i.d. split, but better calibrated.

| Model | Macro F1 Score | Accuracy | ECE | sECE | NLL |
|---|---|---|---|---|---|
| MAP | $0.280 \pm 0.020$ | $0.708 \pm 0.016$ | $0.140 \pm 0.015$ | $-0.140 \pm 0.015$ | $1.514 \pm 0.094$ |
| Deep Ensemble | $0.312 \pm 0.007$ | $0.752 \pm 0.007$ | $0.019 \pm 0.002$ | $\mathbf{-0.015 \pm 0.005}$ | $1.068 \pm 0.016$ |
| MCD | $0.274 \pm 0.024$ | $0.710 \pm 0.021$ | $0.138 \pm 0.013$ | $-0.138 \pm 0.013$ | $1.461 \pm 0.074$ |
| MultiMCD | $0.316 \pm 0.012$ | $\mathbf{0.763 \pm 0.006}$ | $\mathbf{0.014 \pm 0.004}$ | $\mathbf{-0.008 \pm 0.007}$ | $1.026 \pm 0.012$ |
| SWAG | $0.302 \pm 0.009$ | $0.733 \pm 0.005$ | $0.117 \pm 0.006$ | $-0.117 \pm 0.006$ | $1.317 \pm 0.032$ |
| MultiSWAG | $\mathbf{0.337 \pm 0.005}$ | $0.762 \pm 0.001$ | $0.033 \pm 0.001$ | $-0.033 \pm 0.001$ | $\mathbf{1.009 \pm 0.001}$ |
| LL SWAG | $0.294 \pm 0.033$ | $0.721 \pm 0.023$ | $0.104 \pm 0.019$ | $-0.104 \pm 0.020$ | $1.295 \pm 0.091$ |
| LL Laplace | $0.270 \pm 0.010$ | $0.694 \pm 0.015$ | $0.053 \pm 0.015$ | $-0.052 \pm 0.017$ | $1.567 \pm 0.083$ |
| LL MultiLaplace | $0.304 \pm 0.007$ | $0.739 \pm 0.004$ | $0.046 \pm 0.005$ | $0.046 \pm 0.005$ | $1.197 \pm 0.012$ |
| LL BBB | $0.282 \pm 0.011$ | $0.718 \pm 0.009$ | $0.097 \pm 0.006$ | $-0.093 \pm 0.005$ | $1.543 \pm 0.054$ |
| LL MultiBBB | $0.312 \pm 0.008$ | $0.748 \pm 0.002$ | $\mathbf{0.015 \pm 0.003}$ | $-0.012 \pm 0.002$ | $1.164 \pm 0.011$ |
| Rank-1 VI | $0.265 \pm 0.009$ | $0.750 \pm 0.006$ | $0.078 \pm 0.015$ | $-0.076 \pm 0.017$ | $1.198 \pm 0.043$ |
| LL iVON | $0.265 \pm 0.009$ | $0.725 \pm 0.010$ | $0.088 \pm 0.015$ | $-0.084 \pm 0.014$ | $1.331 \pm 0.049$ |
| LL MultiiVON | $0.299 \pm 0.006$ | $\mathbf{0.763 \pm 0.003}$ | $0.019 \pm 0.001$ | $\mathbf{0.011 \pm 0.003}$ | $1.036 \pm 0.006$ |
| SVGD | $0.260 \pm 0.022$ | $0.723 \pm 0.008$ | $0.146 \pm 0.004$ | $-0.146 \pm 0.004$ | $1.619 \pm 0.017$ |
| LL SVGD | $0.265 \pm 0.018$ | $0.737 \pm 0.014$ | $0.118 \pm 0.003$ | $-0.117 \pm 0.003$ | $1.447 \pm 0.045$ |
| SNGP | $0.275 \pm 0.010$ | $0.707 \pm 0.013$ | $0.160 \pm 0.015$ | $-0.160 \pm 0.015$ | $1.459 \pm 0.095$ |

(a) O.o.d. Test Split

| Model | Macro F1 Score | Accuracy | ECE | sECE | NLL |
|---|---|---|---|---|---|
| MAP | $0.460 \pm 0.017$ | $0.813 \pm 0.007$ | $0.104 \pm 0.007$ | $-0.104 \pm 0.007$ | $1.121 \pm 0.087$ |
| Deep Ensemble | $0.308 \pm 0.005$ | $0.752 \pm 0.007$ | $0.020 \pm 0.001$ | $-0.015 \pm 0.005$ | $1.067 \pm 0.012$ |
| MCD | $0.457 \pm 0.010$ | $0.814 \pm 0.002$ | $0.100 \pm 0.011$ | $-0.100 \pm 0.011$ | $1.105 \pm 0.041$ |
| MultiMCD | $0.311 \pm 0.001$ | $0.762 \pm 0.006$ | $\mathbf{0.013 \pm 0.003}$ | $\mathbf{-0.008 \pm 0.006}$ | $1.024 \pm 0.016$ |
| SWAG | $\mathbf{0.491 \pm 0.011}$ | $0.832 \pm 0.003$ | $0.087 \pm 0.002$ | $-0.087 \pm 0.002$ | $0.987 \pm 0.015$ |
| MultiSWAG | $0.333 \pm 0.011$ | $0.761 \pm 0.002$ | $0.033 \pm 0.002$ | $-0.033 \pm 0.002$ | $1.008 \pm 0.002$ |
| LL SWAG | $0.465 \pm 0.043$ | $0.819 \pm 0.016$ | $0.088 \pm 0.012$ | $-0.088 \pm 0.012$ | $1.012 \pm 0.060$ |
| LL Laplace | $0.456 \pm 0.017$ | $0.810 \pm 0.005$ | $0.028 \pm 0.009$ | $-0.026 \pm 0.010$ | $1.045 \pm 0.058$ |
| LL MultiLaplace | $\mathbf{0.489 \pm 0.012}$ | $\mathbf{0.836 \pm 0.001}$ | $0.027 \pm 0.003$ | $0.027 \pm 0.003$ | $\mathbf{0.839 \pm 0.012}$ |
| LL BBB | $0.442 \pm 0.011$ | $0.816 \pm 0.005$ | $0.075 \pm 0.003$ | $-0.075 \pm 0.003$ | $1.143 \pm 0.030$ |
| LL MultiBBB | $0.316 \pm 0.006$ | $0.749 \pm 0.002$ | $\mathbf{0.015 \pm 0.003}$ | $-0.011 \pm 0.003$ | $1.165 \pm 0.009$ |
| Rank-1 VI | $0.442 \pm 0.005$ | $0.819 \pm 0.001$ | $0.082 \pm 0.011$ | $-0.082 \pm 0.011$ | $0.960 \pm 0.052$ |
| LL iVON | $0.447 \pm 0.015$ | $0.812 \pm 0.005$ | $0.076 \pm 0.008$ | $-0.076 \pm 0.008$ | $1.002 \pm 0.028$ |
| LL MultiiVON | $0.294 \pm 0.004$ | $0.763 \pm 0.003$ | $0.019 \pm 0.003$ | $\mathbf{0.010 \pm 0.003}$ | $1.035 \pm 0.005$ |
| SVGD | $0.439 \pm 0.024$ | $0.813 \pm 0.009$ | $0.106 \pm 0.008$ | $-0.105 \pm 0.008$ | $1.303 \pm 0.136$ |
| LL SVGD | $0.453 \pm 0.018$ | $0.822 \pm 0.012$ | $0.094 \pm 0.010$ | $-0.094 \pm 0.009$ | $1.135 \pm 0.234$ |
| SNGP | $0.459 \pm 0.007$ | $0.820 \pm 0.004$ | $0.106 \pm 0.006$ | $-0.106 \pm 0.006$ | $1.081 \pm 0.048$ |

(b) I.d. Validation Split

Table 10: IWILDCAM-WILDS: Detailed results on the evaluation splits. LL = Last-Layer. For the MultiX models, the entire model is ensembled.

### G.3.4 FMOW-WILDS

Following Koh et al. [47], we finetune a DenseNet-121 [35], pretrained on ImageNet [14], for 50 epochs with the Adam optimizer [43] with a batch size of $64$ and a learning rate of $10^{-4}$ that decays by a factor of $0.96$ per epoch. For each model, we replace the linear classification layer of the DenseNet-121 by a randomly initialized one of the appropriate output dimension. iVON uses a prior precision of $100$. We use five seeds per model and build all ensembles by training six models independently and leaving out a different model for each of the five evaluation runs.

We report in the main paper that the Laplace approximation underfits, with a worst-region accuracy of $0.217 \pm 0.012$ and sECE of $-0.583 \pm 0.015$ on the o.o.d. test split. Similarly, MultiLaplace only achieves a worst-region accuracy of $0.301 \pm 0.004$ and sECE of $0.123 \pm 0.004$ on the o.o.d. evaluation split. The accuracy doesn't change when using 100 posterior samples during evaluation, but increases to $0.243$ for 1000 posterior samples. However, using so many samples incurs a significant computational overhead. Note that the better results of Daxberger et al. [13] are most likely due to their usage of models pretrained with ERM. Figure 13 shows additional results for the other models across all regions on the o.o.d. evaluation split, as well as the ECE on the worst region.

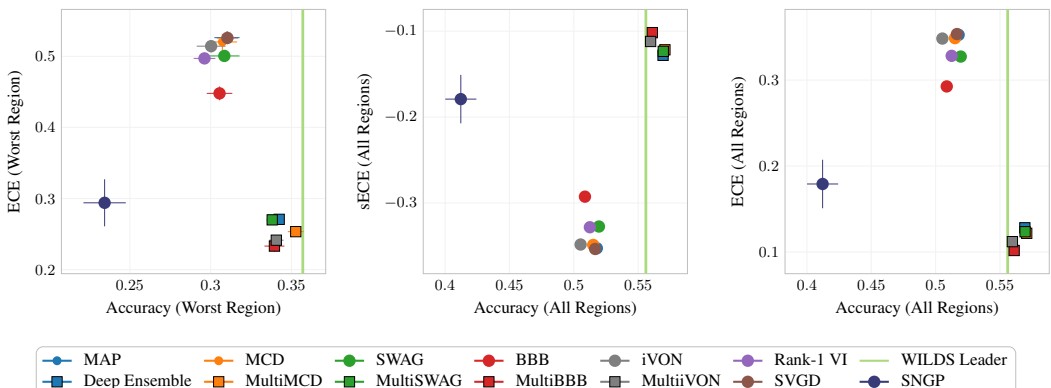

Figure 13: FMoW-WILDS: Accuracy, sECE and ECE on the o.o.d. evaluation split for the region with the lowest accuracy and across all regions (see Figure 2b for accuracy vs. sECE on the worst region). All models are underconfident when evaluated across all regions, but MultiX is less underconfident.

| Model | WR Accuracy | WR ECE | WR sECE | WR NLL | Avg Accuracy | Avg ECE | Avg sECE | Avg NLL |
|---|---|---|---|---|---|---|---|---|
| MAP | $0.310 \pm 0.008$ | $0.526 \pm 0.009$ | $-0.526 \pm 0.009$ | $5.439 \pm 0.117$ | $0.518 \pm 0.003$ | $0.353 \pm 0.002$ | $-0.353 \pm 0.002$ | $3.503 \pm 0.025$ |
| Deep Ensemble | $0.342 \pm 0.003$ | $0.271 \pm 0.004$ | $-0.271 \pm 0.004$ | $3.446 \pm 0.007$ | $0.569 \pm 0.001$ | $0.128 \pm 0.001$ | $-0.128 \pm 0.001$ | $2.141 \pm 0.006$ |
| MCD | $0.307 \pm 0.009$ | $0.520 \pm 0.011$ | $-0.520 \pm 0.011$ | $5.400 \pm 0.118$ | $0.515 \pm 0.002$ | $0.349 \pm 0.004$ | $-0.349 \pm 0.004$ | $3.489 \pm 0.036$ |
| MultiMCD | $\mathbf{0.353 \pm 0.005}$ | $0.253 \pm 0.005$ | $-0.253 \pm 0.005$ | $3.477 \pm 0.030$ | $\mathbf{0.571 \pm 0.000}$ | $0.122 \pm 0.001$ | $-0.122 \pm 0.001$ | $2.150 \pm 0.007$ |
| SWAG | $0.308 \pm 0.009$ | $0.501 \pm 0.007$ | $-0.500 \pm 0.007$ | $4.913 \pm 0.074$ | $0.520 \pm 0.003$ | $0.327 \pm 0.003$ | $-0.327 \pm 0.003$ | $3.150 \pm 0.035$ |
| MultiSWAG | $0.338 \pm 0.003$ | $0.270 \pm 0.003$ | $-0.270 \pm 0.003$ | $3.243 \pm 0.016$ | $0.570 \pm 0.001$ | $0.124 \pm 0.001$ | $-0.124 \pm 0.001$ | $\mathbf{2.016 \pm 0.008}$ |
| LL SWAG | $0.305 \pm 0.005$ | $0.516 \pm 0.003$ | $-0.516 \pm 0.003$ | $5.085 \pm 0.036$ | $0.516 \pm 0.003$ | $0.343 \pm 0.003$ | $-0.343 \pm 0.003$ | $3.271 \pm 0.028$ |
| LL Laplace | $0.212 \pm 0.008$ | $0.590 \pm 0.009$ | $-0.590 \pm 0.009$ | $8.249 \pm 0.435$ | $0.371 \pm 0.014$ | $0.449 \pm 0.012$ | $-0.449 \pm 0.012$ | $5.947 \pm 0.320$ |
| LL Laplace (100 Samples) | $0.213 \pm 0.006$ | $0.588 \pm 0.008$ | $-0.588 \pm 0.008$ | $8.246 \pm 0.355$ | $0.369 \pm 0.012$ | $0.449 \pm 0.010$ | $-0.449 \pm 0.010$ | $5.953 \pm 0.262$ |
| LL MultiLaplace | $0.301 \pm 0.004$ | $\mathbf{0.123 \pm 0.004}$ | $-0.123 \pm 0.004$ | $4.086 \pm 0.047$ | $0.517 \pm 0.002$ | $\mathbf{0.059 \pm 0.002}$ | $\mathbf{0.020 \pm 0.002}$ | $2.744 \pm 0.017$ |
| LL MultiLaplace (100 Samples) | $0.301 \pm 0.003$ | $\mathbf{0.123 \pm 0.003}$ | $-0.123 \pm 0.003$ | $4.088 \pm 0.039$ | $0.517 \pm 0.002$ | $\mathbf{0.059 \pm 0.002}$ | $0.020 \pm 0.002$ | $2.748 \pm 0.016$ |
| LL BBB | $0.306 \pm 0.008$ | $0.448 \pm 0.010$ | $-0.448 \pm 0.010$ | $6.674 \pm 0.343$ | $0.509 \pm 0.003$ | $0.293 \pm 0.003$ | $-0.293 \pm 0.003$ | $4.251 \pm 0.053$ |
| LL MultiBBB | $0.339 \pm 0.006$ | $0.233 \pm 0.008$ | $-0.233 \pm 0.008$ | $4.174 \pm 0.085$ | $0.561 \pm 0.001$ | $0.102 \pm 0.001$ | $-0.102 \pm 0.001$ | $2.617 \pm 0.008$ |
| Rank-1 VI | $0.296 \pm 0.007$ | $0.497 \pm 0.004$ | $-0.497 \pm 0.004$ | $4.645 \pm 0.147$ | $0.512 \pm 0.003$ | $0.328 \pm 0.003$ | $-0.328 \pm 0.003$ | $2.995 \pm 0.037$ |
| LL iVON | $0.300 \pm 0.009$ | $0.514 \pm 0.009$ | $-0.514 \pm 0.009$ | $4.557 \pm 0.112$ | $0.505 \pm 0.003$ | $0.348 \pm 0.002$ | $-0.348 \pm 0.002$ | $3.107 \pm 0.023$ |
| LL MultiiVON | $0.341 \pm 0.004$ | $0.241 \pm 0.004$ | $-0.241 \pm 0.004$ | $\mathbf{3.177 \pm 0.023}$ | $0.560 \pm 0.001$ | $0.112 \pm 0.002$ | $-0.112 \pm 0.002$ | $2.060 \pm 0.009$ |
| SVGD | $0.310 \pm 0.007$ | $0.526 \pm 0.009$ | $-0.526 \pm 0.009$ | $5.542 \pm 0.083$ | $0.517 \pm 0.004$ | $0.354 \pm 0.003$ | $-0.354 \pm 0.003$ | $3.559 \pm 0.041$ |
| SNGP | $0.234 \pm 0.013$ | $0.294 \pm 0.033$ | $-0.294 \pm 0.033$ | $3.419 \pm 0.201$ | $0.412 \pm 0.012$ | $0.179 \pm 0.028$ | $-0.179 \pm 0.028$ | $2.473 \pm 0.126$ |

Table 11: FMoW-WILDS: Detailed results on the o.o.d. evaluation split on the worst region as measured by the accuracy on each region and across all regions. For the MultiX models, the entire model is ensembled, but the single-mode approximation is only applied to the classification head. WR = Worst Region, LL = Last-Layer.

### G.3.5 RxRx1-WILDS

Following Koh et al. [47], we finetune a ResNet-50 [32], pretrained on ImageNet [14], for 90 epochs with the Adam optimizer [43]. For each model, we replace the linear classification layer of the ResNet-50 by a randomly initialized one of the appropriate output dimension. Following Koh et al. [47], we use a learning rate of $10^{-4}$ and weight decay $10^{-5}$. For MCD, we try dropout rates of $0.1$ and $0.2$ and select $0.1$ due to a slightly better accuracy on the evaluation split. iVON uses a prior precision of $100$ as optimized by a grid search. We use five seeds per model and build all ensembles by training six models independently and leaving out a different model for each of the five evaluation runs.

Similar to FMoW, Laplace underperforms accuracy-wise compared to the non-VI algorithms. While we do find a significant increase in accuracy to $0.061 \pm 0.002$ when using 100 posterior samples, Laplace still performs worse than even MAP. However, Laplace is better calibrated with an sECE of $-0.028 \pm 0.001$.

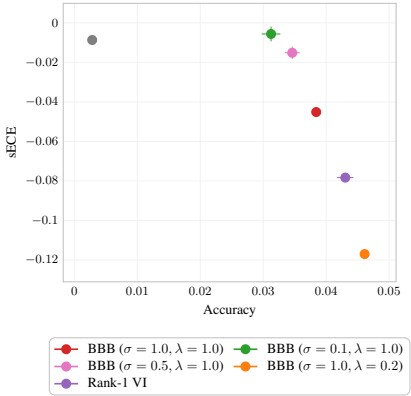

Figure 14: RxRx1-WILDS: BBB and Rank-1 VI under different prior variances $\sigma$ and posterior temperatures $\lambda$. We multiply $\lambda$ to the KL divergence in the ELBO during training to reduce the regularization strength. However, neither small prior variances nor colder posteriors make BBB competitive with the non-VI algorithms.

| Model | i.d. Accuracy | i.d. ECE | i.d. sECE | i.d. NLL | o.o.d. Accuracy | o.o.d. ECE | o.o.d. sECE | o.o.d. NLL |
|---|---|---|---|---|---|---|---|---|
| MAP | $0.105 \pm 0.002$ | $0.232 \pm 0.015$ | $-0.232 \pm 0.015$ | $6.669 \pm 0.121$ | $0.083 \pm 0.001$ | $0.262 \pm 0.015$ | $-0.262 \pm 0.015$ | $7.197 \pm 0.149$ |
| Deep Ensemble | $0.156 \pm 0.001$ | $0.066 \pm 0.001$ | $-0.026 \pm 0.002$ | $\mathbf{5.211 \pm 0.012}$ | $0.122 \pm 0.000$ | $0.071 \pm 0.001$ | $-0.061 \pm 0.002$ | $\mathbf{5.677 \pm 0.019}$ |
| MCD | $0.106 \pm 0.001$ | $0.257 \pm 0.003$ | $-0.257 \pm 0.003$ | $6.924 \pm 0.035$ | $0.083 \pm 0.001$ | $0.288 \pm 0.004$ | $-0.288 \pm 0.004$ | $7.503 \pm 0.043$ |
| MultiMCD | $0.158 \pm 0.001$ | $0.069 \pm 0.001$ | $-0.035 \pm 0.001$ | $5.327 \pm 0.003$ | $0.121 \pm 0.001$ | $0.081 \pm 0.001$ | $-0.073 \pm 0.000$ | $5.836 \pm 0.008$ |
| SWAG | $0.110 \pm 0.001$ | $0.269 \pm 0.009$ | $-0.269 \pm 0.009$ | $6.947 \pm 0.088$ | $0.086 \pm 0.001$ | $0.301 \pm 0.010$ | $-0.301 \pm 0.010$ | $7.549 \pm 0.109$ |
| MultiSWAG | $\mathbf{0.161 \pm 0.001}$ | $0.075 \pm 0.002$ | $-0.042 \pm 0.001$ | $5.299 \pm 0.009$ | $\mathbf{0.126 \pm 0.001}$ | $0.085 \pm 0.001$ | $-0.078 \pm 0.001$ | $5.824 \pm 0.017$ |
| LL Laplace | $0.012 \pm 0.001$ | $0.097 \pm 0.001$ | $-0.097 \pm 0.001$ | $15.280 \pm 0.546$ | $0.010 \pm 0.001$ | $0.099 \pm 0.001$ | $-0.099 \pm 0.001$ | $15.890 \pm 0.579$ |
| LL Laplace (100 Samples) | $0.077 \pm 0.000$ | $0.034 \pm 0.004$ | $-0.011 \pm 0.009$ | $6.624 \pm 0.206$ | $0.061 \pm 0.002$ | $0.037 \pm 0.007$ | $-0.028 \pm 0.007$ | $6.909 \pm 0.271$ |
| LL BBB ($\sigma = 1.0, \lambda = 1.0$) | $0.046 \pm 0.001$ | $0.032 \pm 0.002$ | $-0.032 \pm 0.002$ | $6.657 \pm 0.019$ | $0.038 \pm 0.000$ | $0.045 \pm 0.002$ | $-0.045 \pm 0.002$ | $6.837 \pm 0.012$ |
| LL BBB ($\sigma = 0.5, \lambda = 1.0$) | $0.040 \pm 0.001$ | $\mathbf{0.007 \pm 0.002}$ | $\mathbf{-0.006 \pm 0.003}$ | $6.632 \pm 0.017$ | $0.035 \pm 0.001$ | $0.015 \pm 0.003$ | $-0.015 \pm 0.003$ | $6.737 \pm 0.015$ |
| LL BBB ($\sigma = 0.1, \lambda = 1.0$) | $0.036 \pm 0.002$ | $0.010 \pm 0.001$ | $\mathbf{0.003 \pm 0.004}$ | $6.618 \pm 0.017$ | $0.031 \pm 0.001$ | $\mathbf{0.008 \pm 0.002}$ | $\mathbf{-0.006 \pm 0.004}$ | $6.681 \pm 0.020$ |
| LL BBB ($\sigma = 1.0, \lambda = 0.2$) | $0.054 \pm 0.001$ | $0.102 \pm 0.002$ | $-0.102 \pm 0.002$ | $7.598 \pm 0.044$ | $0.046 \pm 0.001$ | $0.117 \pm 0.002$ | $-0.117 \pm 0.002$ | $7.957 \pm 0.050$ |
| Rank-1 VI | $0.053 \pm 0.001$ | $0.068 \pm 0.001$ | $-0.068 \pm 0.001$ | $7.389 \pm 0.023$ | $0.043 \pm 0.001$ | $0.078 \pm 0.000$ | $-0.078 \pm 0.000$ | $7.577 \pm 0.014$ |
| LL iVON | $0.003 \pm 0.000$ | $\mathbf{0.008 \pm 0.000}$ | $-0.008 \pm 0.000$ | $7.176 \pm 0.012$ | $0.003 \pm 0.000$ | $\mathbf{0.009 \pm 0.001}$ | $-0.009 \pm 0.001$ | $7.213 \pm 0.013$ |
| SVGD | $0.102 \pm 0.001$ | $0.354 \pm 0.005$ | $-0.354 \pm 0.005$ | $8.254 \pm 0.080$ | $0.081 \pm 0.002$ | $0.382 \pm 0.005$ | $-0.382 \pm 0.005$ | $8.936 \pm 0.088$ |
| SNGP | $0.089 \pm 0.006$ | $0.245 \pm 0.023$ | $-0.245 \pm 0.023$ | $6.588 \pm 0.272$ | $0.067 \pm 0.005$ | $0.273 \pm 0.019$ | $-0.273 \pm 0.019$ | $7.070 \pm 0.221$ |

Table 12: RxRx1-WILDS: Detailed results on the i.d. and the o.o.d. evaluation split. LL = Last-Layer. For the MultiX models, the entire model is ensembled, but the single-mode approximation is only applied to the classification head. We evaluate multiple hyperparameter combinations for LL BBB, as the standard parameters do not perform well. The failure of VI is equally present with LL BBB, LL Rank-1 VI, and LL iVON.

### G.3.6 CIVILCOMMENTS-WILDS

We use the pretrained DistilBERT [76] model from HuggingFace transformers [91] with a classification head consisting of two linear layers with a ReLU nonlinearity and a Dropout unit with a drop rate of 0.2 between them. Following Koh et al. [47], we finetune the pretrained checkpoint with a learning rate of $1 \cdot 10^{-5}$ and, where applicable, a weight decay factor of $1 \cdot 10^{-2}$ for three epochs using the Adam optimizer [43]. SWAG collects ten parameter samples during the last two epochs of training. iVON uses a prior precision of 500, as optimized by a grid search. We use five seeds for all non-ensembled models. The ensembles are build from four of the five single-model versions, leaving out a different member per model to create five different ensembled models of four members each.

We note in the main paper that MCD results in less accurate and more overconfident models. We investigate this further by experimenting with different dropout rates in Figure 15. While a dropout rate of 0.1 had no impact, dropout rates of 0.05 and 0.01 lead to progressively better accuracy and calibration, coming close to MAP. However, there is still no accuracy or calibration benefit to be gained from using MCD.

### G.3.7 AMAZON-WILDS

We use the pretrained DistilBERT [76] model from HuggingFace transformers [91] with a classification head consisting of two linear layers with a ReLU nonlinearity and a Dropout unit with a drop rate of 0.2 between them. Following Koh et al. [47], we finetune the pretrained checkpoint with a learning rate of $10^{-5}$ and, where applicable, a weight decay factor of $10^{-2}$ using the Adam optimizer [43]. Contrary to Koh et al. [47], we finetune for five epochs, as we find that the validation accuracy

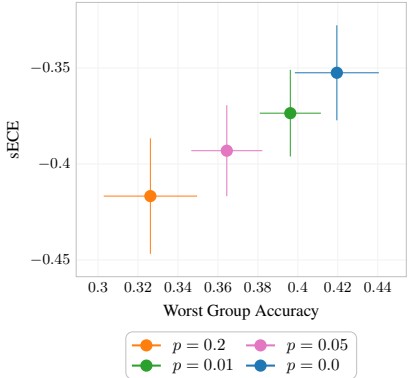

Figure 15: CIVILCOMMENTS-WILDS: Accuracy and sECE for different MCD dropout rates $p$. While smaller dropout rates improve the accuracy, the models are still less accurate and more overconfident than MAP.

| Model | WG Accuracy | WG ECE | WG sECE | WG NLL | Avg Accuracy | Avg ECE | Avg sECE | Avg NLL |
|---|---|---|---|---|---|---|---|---|
| MAP | $0.420 \pm 0.021$ | $0.353 \pm 0.025$ | $-0.353 \pm 0.025$ | $1.455 \pm 0.086$ | $0.916 \pm 0.001$ | $0.012 \pm 0.003$ | $-0.012 \pm 0.003$ | $0.207 \pm 0.001$ |
| Deep Ensemble | $0.419 \pm 0.008$ | $0.349 \pm 0.010$ | $-0.349 \pm 0.010$ | $1.416 \pm 0.032$ | $0.916 \pm 0.000$ | $0.010 \pm 0.001$ | $-0.010 \pm 0.001$ | $0.204 \pm 0.000$ |
| MCD ($p = 0.2$) | $0.326 \pm 0.023$ | $0.417 \pm 0.030$ | $-0.417 \pm 0.030$ | $1.391 \pm 0.074$ | $0.918 \pm 0.000$ | $0.007 \pm 0.005$ | $0.006 \pm 0.006$ | $0.204 \pm 0.001$ |
| MCD ($p = 0.1$) | $0.325 \pm 0.021$ | $0.418 \pm 0.027$ | $-0.418 \pm 0.027$ | $1.390 \pm 0.070$ | $0.918 \pm 0.000$ | $\mathbf{0.007 \pm 0.005}$ | $\mathbf{0.006 \pm 0.005}$ | $0.204 \pm 0.001$ |
| MCD ($p = 0.05$) | $0.364 \pm 0.018$ | $0.393 \pm 0.024$ | $-0.393 \pm 0.024$ | $1.430 \pm 0.065$ | $0.918 \pm 0.000$ | $\mathbf{0.005 \pm 0.002}$ | $-0.003 \pm 0.004$ | $\mathbf{0.203 \pm 0.001}$ |
| MCD ($p = 0.01$) | $0.396 \pm 0.015$ | $0.374 \pm 0.023$ | $-0.374 \pm 0.023$ | $1.452 \pm 0.114$ | $0.917 \pm 0.000$ | $0.011 \pm 0.003$ | $-0.011 \pm 0.003$ | $0.206 \pm 0.001$ |
| MultiMCD ($p = 0.2$) | $0.326 \pm 0.005$ | $0.412 \pm 0.007$ | $-0.412 \pm 0.007$ | $1.363 \pm 0.022$ | $0.919 \pm 0.000$ | $0.009 \pm 0.002$ | $0.009 \pm 0.002$ | $\mathbf{0.203 \pm 0.000}$ |
| SWAG | $0.448 \pm 0.021$ | $\mathbf{0.197 \pm 0.041}$ | $\mathbf{-0.184 \pm 0.027}$ | $0.872 \pm 0.050$ | $0.877 \pm 0.024$ | $0.152 \pm 0.024$ | $0.152 \pm 0.024$ | $0.396 \pm 0.019$ |
| MultiSWAG | $0.429 \pm 0.016$ | $\mathbf{0.183 \pm 0.018}$ | $\mathbf{-0.183 \pm 0.018}$ | $\mathbf{0.819 \pm 0.011}$ | $0.901 \pm 0.002$ | $0.184 \pm 0.002$ | $0.184 \pm 0.002$ | $0.388 \pm 0.004$ |
| LL Laplace | $0.424 \pm 0.016$ | $0.348 \pm 0.018$ | $-0.347 \pm 0.018$ | $1.438 \pm 0.065$ | $0.916 \pm 0.001$ | $0.011 \pm 0.002$ | $-0.011 \pm 0.002$ | $0.207 \pm 0.001$ |
| LL MultiLaplace | $0.420 \pm 0.008$ | $0.348 \pm 0.010$ | $-0.348 \pm 0.010$ | $1.411 \pm 0.032$ | $0.916 \pm 0.000$ | $0.010 \pm 0.001$ | $-0.009 \pm 0.001$ | $0.204 \pm 0.000$ |
| LL BBB | $\mathbf{0.537 \pm 0.032}$ | $0.362 \pm 0.032$ | $-0.361 \pm 0.033$ | $2.192 \pm 0.278$ | $0.918 \pm 0.002$ | $0.056 \pm 0.002$ | $-0.056 \pm 0.002$ | $0.333 \pm 0.017$ |
| LL MultiBBB | $\mathbf{0.525 \pm 0.012}$ | $0.338 \pm 0.012$ | $-0.338 \pm 0.012$ | $1.801 \pm 0.078$ | $0.922 \pm 0.000$ | $0.041 \pm 0.001$ | $-0.041 \pm 0.001$ | $0.265 \pm 0.003$ |
| Rank-1 VI | $\mathbf{0.540 \pm 0.028}$ | $0.373 \pm 0.030$ | $-0.373 \pm 0.030$ | $2.065 \pm 0.179$ | $0.917 \pm 0.002$ | $0.060 \pm 0.002$ | $-0.060 \pm 0.002$ | $0.319 \pm 0.007$ |
| LL iVON | $0.480 \pm 0.045$ | $0.421 \pm 0.048$ | $-0.421 \pm 0.048$ | $2.198 \pm 0.263$ | $0.919 \pm 0.003$ | $0.054 \pm 0.002$ | $-0.054 \pm 0.002$ | $0.299 \pm 0.014$ |
| LL MultiiVON | $0.465 \pm 0.011$ | $0.396 \pm 0.015$ | $-0.396 \pm 0.015$ | $1.752 \pm 0.073$ | $\mathbf{0.924 \pm 0.001}$ | $0.039 \pm 0.001$ | $-0.039 \pm 0.001$ | $0.240 \pm 0.002$ |
| SVGD | $0.384 \pm 0.068$ | $0.380 \pm 0.079$ | $-0.379 \pm 0.079$ | $1.393 \pm 0.154$ | $0.915 \pm 0.003$ | $0.011 \pm 0.005$ | $-0.008 \pm 0.008$ | $0.208 \pm 0.002$ |
| SNGP | $0.394 \pm 0.039$ | $0.388 \pm 0.036$ | $-0.388 \pm 0.036$ | $1.341 \pm 0.078$ | $0.919 \pm 0.001$ | $0.014 \pm 0.006$ | $-0.014 \pm 0.006$ | $0.206 \pm 0.004$ |

Figure 16: CIVILCOMMENTS-WILDS: Detailed results on the o.o.d. evaluation split for the worst group (WG, determined by the accuracy on each group) and averaged over all groups. LL = Last-Layer.

is still increasing after three epochs. SWAG collects 30 parameter samples during the last two epochs of training. We also experiment with last-layer versions of SWAG and MCD, but find both to perform very similar to MAP (see Table 13). iVON uses a prior precision of $500$, as optimized by a grid search. We use six seeds for all non-ensembled models. The ensembles are build from five of the six single-model versions, leaving out a different member per model to create five different ensembled models of five members each.

| Model | o.o.d. 10 Accuracy | o.o.d. Accuracy | o.o.d. ECE | o.o.d. sECE | o.o.d. NLL | i.d. 10 Accuracy | i.d. Avg Accuracy | i.d. ECE | i.d. sECE | i.d. NLL |
|---|---|---|---|---|---|---|---|---|---|---|
| MAP | $0.453 \pm 0.010$ | $0.655 \pm 0.003$ | $0.067 \pm 0.006$ | $-0.067 \pm 0.006$ | $0.815 \pm 0.007$ | $0.477 \pm 0.008$ | $0.678 \pm 0.002$ | $0.049 \pm 0.007$ | $-0.049 \pm 0.007$ | $0.755 \pm 0.005$ |
| Deep Ensemble | $0.453 \pm 0.000$ | $0.659 \pm 0.001$ | $0.058 \pm 0.002$ | $-0.058 \pm 0.002$ | $0.800 \pm 0.001$ | $0.480 \pm 0.000$ | $0.682 \pm 0.000$ | $0.040 \pm 0.002$ | $-0.040 \pm 0.002$ | $0.742 \pm 0.001$ |
| MCD | $0.447 \pm 0.013$ | $0.657 \pm 0.002$ | $0.020 \pm 0.011$ | $-0.019 \pm 0.012$ | $0.789 \pm 0.004$ | $0.472 \pm 0.012$ | $0.678 \pm 0.001$ | $0.015 \pm 0.006$ | $-0.002 \pm 0.013$ | $0.741 \pm 0.003$ |
| MultiMCD | $0.451 \pm 0.005$ | $0.660 \pm 0.001$ | $0.012 \pm 0.003$ | $-0.012 \pm 0.003$ | $0.780 \pm 0.001$ | $0.475 \pm 0.007$ | $0.682 \pm 0.000$ | $0.007 \pm 0.002$ | $0.005 \pm 0.003$ | $0.733 \pm 0.000$ |
| LL MCD | $0.451 \pm 0.008$ | $0.656 \pm 0.003$ | $0.069 \pm 0.008$ | $-0.069 \pm 0.008$ | $0.816 \pm 0.011$ | $0.478 \pm 0.011$ | $0.679 \pm 0.002$ | $0.051 \pm 0.009$ | $-0.051 \pm 0.009$ | $0.756 \pm 0.009$ |
| SWAG | $0.436 \pm 0.011$ | $0.639 \pm 0.006$ | $0.032 \pm 0.003$ | $0.031 \pm 0.004$ | $0.840 \pm 0.014$ | $0.460 \pm 0.010$ | $0.658 \pm 0.006$ | $0.047 \pm 0.004$ | $0.047 \pm 0.004$ | $0.807 \pm 0.015$ |
| MultiSWAG | $0.443 \pm 0.005$ | $0.646 \pm 0.001$ | $0.040 \pm 0.001$ | $0.040 \pm 0.001$ | $0.828 \pm 0.002$ | $0.469 \pm 0.005$ | $0.667 \pm 0.001$ | $0.057 \pm 0.001$ | $0.057 \pm 0.001$ | $0.796 \pm 0.003$ |
| LL SWAG | $0.452 \pm 0.012$ | $0.656 \pm 0.003$ | $0.048 \pm 0.009$ | $-0.048 \pm 0.009$ | $0.802 \pm 0.006$ | $0.474 \pm 0.010$ | $0.679 \pm 0.002$ | $0.031 \pm 0.008$ | $-0.030 \pm 0.009$ | $0.747 \pm 0.005$ |
| LL Laplace | $0.455 \pm 0.009$ | $0.654 \pm 0.003$ | $0.067 \pm 0.006$ | $-0.067 \pm 0.006$ | $0.816 \pm 0.006$ | $0.482 \pm 0.009$ | $0.678 \pm 0.002$ | $0.048 \pm 0.007$ | $-0.048 \pm 0.007$ | $0.756 \pm 0.004$ |
| LL MultiLaplace | $0.453 \pm 0.000$ | $0.659 \pm 0.001$ | $0.058 \pm 0.001$ | $-0.058 \pm 0.001$ | $0.800 \pm 0.001$ | $0.480 \pm 0.000$ | $0.682 \pm 0.000$ | $0.040 \pm 0.002$ | $-0.040 \pm 0.002$ | $0.742 \pm 0.001$ |
| LL BBB | $0.527 \pm 0.006$ | $0.695 \pm 0.007$ | $0.154 \pm 0.005$ | $-0.154 \pm 0.005$ | $0.898 \pm 0.019$ | $0.560 \pm 0.000$ | $0.730 \pm 0.006$ | $0.128 \pm 0.004$ | $-0.128 \pm 0.004$ | $0.778 \pm 0.015$ |
| LL MultiBBB | $\mathbf{0.533 \pm 0.000}$ | $\mathbf{0.709 \pm 0.002}$ | $0.105 \pm 0.001$ | $-0.105 \pm 0.001$ | $\mathbf{0.748 \pm 0.001}$ | $\mathbf{0.573 \pm 0.000}$ | $\mathbf{0.746 \pm 0.001}$ | $0.079 \pm 0.002$ | $-0.079 \pm 0.002$ | $\mathbf{0.648 \pm 0.002}$ |
| Rank-1 VI | $0.527 \pm 0.005$ | $0.695 \pm 0.003$ | $0.173 \pm 0.004$ | $-0.173 \pm 0.004$ | $0.923 \pm 0.015$ | $0.558 \pm 0.004$ | $0.729 \pm 0.003$ | $0.147 \pm 0.004$ | $-0.147 \pm 0.004$ | $0.801 \pm 0.010$ |
| LL iVON | $0.458 \pm 0.010$ | $0.661 \pm 0.002$ | $0.053 \pm 0.010$ | $-0.053 \pm 0.010$ | $0.794 \pm 0.008$ | $0.484 \pm 0.009$ | $0.684 \pm 0.002$ | $0.037 \pm 0.010$ | $-0.037 \pm 0.010$ | $0.737 \pm 0.006$ |
| LL MultiiVON | $0.459 \pm 0.007$ | $0.665 \pm 0.001$ | $0.045 \pm 0.002$ | $-0.045 \pm 0.002$ | $0.779 \pm 0.001$ | $0.484 \pm 0.005$ | $0.687 \pm 0.001$ | $0.029 \pm 0.003$ | $-0.029 \pm 0.002$ | $0.724 \pm 0.001$ |
| SVGD | $0.456 \pm 0.010$ | $0.661 \pm 0.003$ | $0.049 \pm 0.005$ | $-0.049 \pm 0.005$ | $0.793 \pm 0.011$ | $0.477 \pm 0.010$ | $0.682 \pm 0.002$ | $0.035 \pm 0.005$ | $-0.034 \pm 0.005$ | $0.740 \pm 0.008$ |
| SNGP | $0.451 \pm 0.013$ | $0.661 \pm 0.001$ | $0.053 \pm 0.008$ | $-0.053 \pm 0.008$ | $0.800 \pm 0.002$ | $0.487 \pm 0.013$ | $0.685 \pm 0.001$ | $0.036 \pm 0.008$ | $-0.035 \pm 0.008$ | $0.737 \pm 0.001$ |

Table 13: AMAZON-WILDS: Detailed results on the i.d. and the o.o.d. evaluation splits. LL = Last-Layer.

