# OpenReview forum: "Beyond Deep Ensembles: A Large-Scale Evaluation of Bayesian Deep Learning under Distribution Shift"
_NeurIPS.cc/2023/Conference — NeurIPS 2023 poster_

### Official Review · Reviewer_eT5S · 2023-06-20

**Soundness:** 2 fair
**Presentation:** 1 poor
**Contribution:** 1 poor
**Rating:** 4
**Confidence:** 4

**Summary:**

This paper evaluates several BDL algorithm on real-world datasets (and real-world distribution shifts) in terms of generalisation (accuracy/correlation metrics) and uncertainty estimation quality (calibration/signed calibration).

**Strengths:**

1. I think there is a need for benchmarking BDL techniques, and the paper tries to tackle that problem.

2. The point about batch normalisation leading to really poor o.o.d. data performance is interesting, and makes sense. This could be emphasised more as a contribution of the work (if it is not already known, I'm not sure!).

**Weaknesses:**

Overall, I vote for rejection. I am not convinced that this paper is of interest to the community, and ready for publication yet. However, I am open to changing my mind if I see evidence from the authors. I've included suggestions and questions for the authors that would enable me to raise my score. It's possible I misunderstood key details; I am more than happy to be corrected.

1. I do not think the paper makes a substantial enough contribution to be accepted at NeurIPS. The contribution of the signed calibration metric is nice but small. If there was a clear takeaway recommendation from the experiment results, that would be a useful contribution to the community, but as I can understand, there isn't one. Despite this, the experiment section is mostly descriptive and does not provide greater insight and understanding about when one method is better than another. Either making the argument that the experiment section makes clearer and using the results to provide greater insight and understanding would help me increase my score. For example, the result that Bayes-by-Backprop performs very well here is interesting and surprising to me. Why is this the case? VI usually underfits, so does VI regularise to the base model more strongly? Which would be good in the pre-trained setup? This result contradicts other work ([1]) where VI often performs poorly and underfits. Please explain the contradiction.

2. Missing reference to Nado et. al [1]. The need for baselines is clear, but what does this work offer over this work? I'm quite surprised that I didn't see the reference in the paper. For what it's worth, I think the Nado et al. baselines are also limited (only every evaluating marignal uncertainty, nothing about sequential decision making), so I think good work remains to be in this domain, but I don't think the paper offers much over this work. Would you please explain to me, in clear and precise terms, the contribution of this submission relative to this work?

3. Missing baselines: give the point of this paper is the comparison (at least), it is important to be comprehensive in terms of the baselines covered. I would like to see SNGP, DUQ, and some frequentist approaches like temperature rescaling. This is a clear weakness of the paper. Further, not including function-space approaches is a major limitation: the paper claims to do a thorough investigation, and leaves out an important class of methods.

4. A key point of the paper is that existing evaluations e.g., on CIFAR-10-C are unrealistic because the distribution shift is constructed and not real world. I think this point is interesting, but I would like to see evidence to support it. For example, if one would make different (and importantly different) conclusions based on benchmarking on CIFAR-10-C vs on WILDS. Further, other BDL papers do look at WILDS datasets e.g. [4], in case you weren't aware of this. Evaluating on WILDS is not sufficient for novelty, in my view.

5. "modern single-model BDL algorithms approximate the parameter posterior better than deep ensembles ..." Well, for one, your evaluation looks at total variation in predictive space, so you can make no claim about the __parameter__ posterior. And, furthermore, recent work [5] shows that the HMC chains from the Izmailov et al paper have not converged well, so I don't think a claim about "approximating the posterior" is valid, you can only claim to approximate the HMC chains. And since the HMC chains perform badly out of distribution, it is not clear that lower total variation is better! See [5] on that, which argues that we might not want to be doing full network inference in the first place.

6. The writing of the experiment section is very poor. The figures aren't referenced in the text, the writing is mostly around the results, but offers little in terms of understanding, insight, conclusions, or discussion. I found this very hard to read, and I am left confused: what are your take away messages? What is the contribution of the work?

7. A claim is made "BDL is in many cases competitive with algorithms that are specifically designed for OOD generalisation" (Line 393,394). This does not seem to be supported: what benchmarks did you compare to that are specifically designed for OOD data?


Minor points:
- "typical deep neural networks are highly confident on o.o.d. data". I'd argue this is a bit different with large pretrained models.
- Since your results used pre-trained model, [6, 7] are relevant citations because they justify pre-training in a Bayesian setting.

[1] Nado, Zachary, et al. "Uncertainty Baselines: Benchmarks for uncertainty & robustness in deep learning." arXiv preprint arXiv:2106.04015 (2021).

[2] Liu, Jeremiah, et al. "Simple and principled uncertainty estimation with deterministic deep learning via distance awareness." Advances in Neural Information Processing Systems 33 (2020): 7498-7512.

[3] Van Amersfoort, Joost, et al. "Uncertainty estimation using a single deep deterministic neural network." International conference on machine learning. PMLR, 2020.

[4] Daxberger, Erik, et al. "Laplace redux-effortless bayesian deep learning." Advances in Neural Information Processing Systems 34 (2021): 20089-20103.

[5] Sharma, Mrinank, et al. "Do Bayesian Neural Networks Need To Be Fully Stochastic?." International Conference on Artificial Intelligence and Statistics. PMLR, 2023.

[6] Shwartz-Ziv, Ravid, et al. "Pre-train your loss: Easy bayesian transfer learning with informative priors." Advances in Neural Information Processing Systems 35 (2022): 27706-27715.

[7] Sharma, Mrinank, et al. "Incorporating Unlabelled Data into Bayesian Neural Networks." arXiv preprint arXiv:2304.01762 (2023).

**Questions:**

See also the weaknesses for the list of questions. Some additional questions are:

1. What are the differences between WILDS and e.g., CIFAR-10, other than the "realistic" dataset shift? Would you be more satisfied with BDL people evaluating on Imagenet-C?

2. How are hyperparameters selected?

3. How do you tune the prior precision for the Laplace approximation? In my experience, it can be quite difficult to tune well. I tried to look in the appendix, but it said that I didn't have permission to view the file.

4. Line 328 says Bayes-by-Backprop has a "last-layer nature". Why? People perform BBB on the full network.

**Limitations:**

Seems fine to me; see weaknesses.

---

> ### Author Rebuttal · Authors · 2023-08-09
>
> Thank you for your effort in reviewing our paper. We are happy that you found our results interesting and that you think there is a need for benchmarking BDL techniques. We will address your concerns regarding our evaluation below and refer to our global response, which summarizes our takeaway messages, presents preliminary results for SNGP, discusses our algorithm selection in detail, and provides the list of citations.
>
> Overall, we believe our results are highly relevant to the BDL community since we compare many current SOTA algorithms (MultiX, natural gradient VI) on a diverse range of realistic tasks (finetuning, transformer-based models, large-scale regression) and find surprising results (cf. our takeaway messages). Our results allow a direct comparison with the non-Bayesian OOD generalization baselines of [2]. Reviewers Sjhm, 1hhr, hcj9, and CBPk agree with us on the potential impact of the results.
>
> **Batch normalization.** We discuss this effect in more detail in Appendix E and refer to [1] which demonstrates the problem with Batch Normalization on more datasets.
>
> **Size of the contribution.** We agree that clear and actionable takeaway messages are very useful and will add them, as stated in the global response, to our revised conclusion. As we wanted to focus on the evaluation of a wide range of methods on many diverse datasets, we decided against investigating a single phenomenon in depth. This approach follows similar large-scale evaluations published at NeurIPS both within BDL [5,8] and other domains [14,15] (see the references below). Further, NeurIPS 2023 explicitly invites "Evaluation (e.g., methodology, meta studies, replicability and validity)" submissions.
>
> **Performance of BBB.** Please note that the performance of BBB on the image classification tasks is in line with the results of [2] on similar tasks. Further, the regularization of MAP on the text classification tasks is strong (weight decay of ${10^{-2}}$, as mandated by [3]). Reducing the weight decay makes MAP more accurate on CivilComments, but still less accurate than BBB. As shown in Figure 3 in the uploaded PDF, reducing the regularization of BBB by introducing a tempering factor does not significantly change BBB's accuracy. These results indicate that BBB's performance is at most partially due to a difference in regularization, and otherwise represents an actual benefit of performing mean-field VI. We will add an extended discussion of this topic to the paper revision.
>
> **Distinction to Nado et al.** We thank you for the suggestion to cite [2]. We added a reference and short summary in the related work section. We believe that our paper offers a number of important new results compared to their work:
>
> - A systematic evaluation of ensembles of probabilistic single-mode posterior approximations (MultiX). Reviewers hcj9 and 1hhR agree with us that this evaluation is highly relevant. Further, we include approaches not considered in [2], e.g., the Laplace approximation and iVON (natural gradient VI).
> - A systematic evaluation of BDL algorithms for the finetuning of pre-trained models.
> - The inclusion of a large-scale regression task. To the best of our knowledge, such an evaluation is yet completely missing from the BDL literature, including [2], who only evaluate on a number of small UCI datasets.
> - Insights into the posterior approximation quality by comparing to the HMC samples from [6].
> - Strict adherence to the evaluation protocol of [3], which allows for a fair comparison with the non-Bayesian baselines on the WILDS leaderboard.
>
> **Missing baselines.** Following your suggestion, we add a comparison with SNGP during the revision. We include preliminary results in the uploaded PDF file (Figure 1a). Furthermore, we discuss our algorithm selection in the global response and add this discussion to the appendix.
>
> **Realism of evaluations.** We want to clarify that we do not think that evaluations on CIFAR-10-C are unrealistic, but one of our main research questions is whether the conclusions on these artificial distribution shifts transfer to real-world distribution shifts. Regardless of our concrete results, we see this as an important point by itself, as it has direct implications for the applicability of BDL. We do find novel and surprising results (c.f. takeaway messages). Please also note that while [9] evaluates on WILDS, they finetune models from [2], originally trained with algorithms that are specially designed for domain generalization. This prevents a fair comparison of the algorithms' performance in isolation. Further, they consider a significantly smaller number of algorithms.
>
> **Comparison with HMC.** We changed the formulations in the paper to be more cautious about the implications in parameter space. We now refer to [7] to acknowledge the problems with the HMC samples, but argue that they currently still provide the best available way to assess the posterior approximation quality on a complex task.
>
> **Competitiveness with special OOD generalization algorithms.** We added an explicit reference to the WILDS leaderboard [3] which provides an extensive selection of OOD generalization algorithms.
>
> **Overconfidence of large pretrained models.** We agree that such issues are less severe on large models, yet, our results for MAP on FMoW (Figure 3b) and CivilComments (Figure 4) show that these models can still be highly overconfident on OOD data.
>
> **Hyperparameters.** We selected hyperparameters based on a combination of grid search and default values, and tuned the prior precision of Laplace based on the marginal likelihood of the training data.
>
> Finally, we appreciate your suggestions regarding our writing style and additional citations and will include this feedback in our revision.
>
> [14] Veilleux, Olivier, et al. "Realistic evaluation of transductive few-shot learning" NeurIPS 2021
>
> [15] Setlur, Amrith, et al. "Two sides of meta-learning evaluation: In vs. out of distribution" NeurIPS 2021

---

> > ### Comment · Reviewer_eT5S · 2023-08-13
> >
> > Thanks for the comments. They make sense. Overall, I have doubts over the contribution of the paper still, and how interesting the findings are. I raise my score to borderline reject.

---

> > > ### Author Response · Authors · 2023-08-14
> > >
> > > Thank you very much for your response and for raising the score! We are interested in details about the concerns that were not addressed by our rebuttal. We would be grateful if you could elaborate a bit more so that we have the chance to further improve our paper. In particular, if you think any specific additional experiments would strengthen the paper, please let us know.

---

### Official Review · Reviewer_hcj9 · 2023-06-23

**Soundness:** 3 good
**Presentation:** 4 excellent
**Contribution:** 3 good
**Rating:** 6
**Confidence:** 4

**Summary:**

This paper presents an empirical comparison between various Bayesian methods/approaches on out-of-distribution (OOD) data. The authors focus on non-MCMC based methods (such as Bayes By Backprop, SWAG, and Laplace approximation). Methods are compared on challenging OOD benchmarks from the WILDS collection and using several network architectures. All compared methods are evaluated in two flavours, first as a single-mode posterior and second in an ensemble of models. Results are somewhat inconclusive, different methods tend to work better in terms of calibration and generalization on different datasets.

**Strengths:**

* I think that the paper makes a nice contribution to the community. It may be hard to pick a model that perform well on OOD data in the Bayesian model zoo, and this paper makes another step towards a better understanding of this question.
* These types of projects are hard to manage. There are many choices to be made in terms of methods, datasets, evaluation metrics, etc. I think that the authors made good choices in all of these aspects. For instance, they picked methods that reflect typical Bayesian methods people use, and they focused on 2-3 evaluation metrics that capture both uncertainty quantification and generalization capabilities adequately.
* For the most part, the paper is written well. The authors justify many of their choices.
* Evaluating the ensemble version of single-mode-posteriors is a good idea and was mandatory in my opinion.
* The results seem reproducible, fully experimental details were given and the code was provided.

**Weaknesses:**

* I find it hard to understand the key takeaways. I think the authors should present the main conclusion from each experiment and the key takeaways from all experiments (e.g., inside a text box with some background color, or in bold).
* As the results are inconclusive, e.g.,  Rank-1 VI is best calibrated on PovertyMAP, but less so on IWILDCAM, FMOW, and RXRX1, I believe it would have been beneficial to suggest a possible explanation (or perform an empirical analysis) for the reasons that make a method work better on some datasets and worse on others.
* An important evaluation metric that is missing in my opinion is in terms of computational complexity (memory and time of each method). It may be important information when comparing the methods as well.
* The authors evaluate last-layer Bayesian methods, which is important and great. An important method that I find missing is deep kernel learning [1, 2]. It is also a popular last-layer Bayesian method and, from my experience, it tends to work better than most of the compared methods.
* In terms of exposition:
  * I think that the authors should explain at the beginning of Section 5 or 5.1 that the analyses of the results are given in Sections 5.3 and 5.4. It is not clear until reaching there.
  * In the figures, the font of the tick labels, axis labels, and method names should be larger.

[1]  Wilson, A. G., Hu, Z., Salakhutdinov, R., & Xing, E. P. (2016, May). Deep kernel learning. In Artificial intelligence and statistics (pp. 370-378). PMLR.

[2]  Calandra, R., Peters, J., Rasmussen, C. E., & Deisenroth, M. P. (2016, July). Manifold Gaussian processes for regression. In 2016 International joint conference on neural networks (IJCNN) (pp. 3338-3345). IEEE.

**Questions:**

* I may be wrong, but from the main text it seems that you evaluated BBB on the layer only, why not use it on the full network?
* The reference format is a bit odd. For instance, not all authors names are written. Please check that you adhere to NeurIPS guidelines.

**Limitations:**

The authors addressed limitations of their work.

---

> ### Author Rebuttal · Authors · 2023-08-09
>
> Thank you very much for your overall positive remarks. We are happy that you think "that the paper makes a nice contribution to the community" and that we made "good choices" regarding methods, datasets, and metrics. We address your remaining questions below. Please also see the global response for a list of takeaway messages, results for SNGP, and a list of all citations.
>
> **Key takeaways.** We added an itemized list of takeaway messages as part of the conclusion. Please see our global response for the takeaway message.
>
> **Explanation of performance differences.** Please note that our results are largely conclusive within groups of similar tasks, e.g. the image classification tasks iWildCam, FMoW, and RxRx1, and the text classification tasks CivilComments and Amazon. For Rank-1 VI in particular, we strongly suspect that the good performance on CIFAR (classification) and PovertyMap (regression) and the bad performance on the other datasets indicates that the multi-modal rank-1 factors are only sufficiently multi-modal when performing inference on the full network, but not when performing inference on only the last layers. Following your suggestions, we add a more clear discussion of this point in the revised manuscript.
>
> **Computational complexity.** We agree that computational complexity is an important metric. Please note that we report the runtime of each method on each dataset in the appendix (Section F, Tables 1 and 2). We will add a similar comparison for the memory consumption as given below:
>
>
> | Method     | Memory consumption relative to MAP                                 |
> |------------|--------------------------------------------------------------------|
> | MCD        | 1                                                                  |
> | SWAG       | 1                                                                  |
> | LL Laplace | $\sim 1$                                                           |
> | Ensembles  | 1 (since the models can be trained independently)                  |
> | BBB        | 2                                                                  |
> | Rank-1 VI  | $\sim 1$ + #components $\cdot \sqrt{\text{parameter count}}$ |
> | SVGD       | #particles                                                      |
> | iVON       | $\sim 2$, but no additional memory overhead due to an optimizer                                                      |
>
> The memory overhead is of course lower when using the algorithms only on the last layer(s).
>
> **Deep Kernel Learning.** Thank you for the suggestion. We added SNGP, a scalable variant of DKL, as an additional baseline. Please see our global response for further details and a general remark regarding our algorithm selection.
>
> **Overview of Section 5.** Thank you very much for the suggestion. We added a short overview over the section as well as references to the subsections at the beginning of Section 5.
>
> **Font size in figures.** Again, thank you very much for the suggestion. We will take it into account for the next revision of the paper.
>
> **Last-layer BBB.** On CIFAR-10-(C) and PovertyMap, we trained BBB on the full network. On the other tasks from WILDS, we trained BBB only on the classification heads of the respective models. This was mainly done to limit BBB's runtime, which is of major concern given its computational overhead.
>
> **Reference format.** Thank you very much for the notice. We will check all references in the paper and ensure that they comply with the NeurIPS guidelines in our revision.

---

> > ### Comment · Reviewer_hcj9 · 2023-08-13
> > **Response to Authors**
> >
> > I thank the authors for the response. I believe the authors addressed some of the concerns well (including mine) while less so to others, such as the need for a proper analysis as to why and when a certain method is preferred (which was also raised by Reviewer eT5S). Overall, I believe that this paper makes a nice contribution to the community and therefore I retain my original score.

---

> > > ### Author Response · Authors · 2023-08-14
> > >
> > > Thank you very much for your response. We are happy that you find our paper to make a "nice contribution to the community".

---

### Official Review · Reviewer_Sjhm · 2023-07-05

**Soundness:** 4 excellent
**Presentation:** 4 excellent
**Contribution:** 3 good
**Rating:** 8
**Confidence:** 4

**Summary:**

This paper presents a systematic comprehensive evaluation of a wide range of scalable Bayesian deep learning (BDL) algorithms in distribution-shited real-world data scenarios. A signed version of calibration error is presented, which allows for the identification of overconfidence and underconfidence rather than only absolute calibration. The study bolsters the ongoing effort in the BDL community to establish the practicality of BDL algorithms in safety-critical settings.

**Strengths:**

In my opinion, this line of work has huge potential but is in dire need of a "reality check" in order to make tangible, impactful progress in real-world safety-critical settings. I believe this paper makes real progress in that direction. Indeed, the primary motivating statement behind uncertainty estimation is one which appeals to its utility in safety-critical applications, i.e. how and when can we trust our predictions. However, perhaps ironically, much emphasis has been placed on reiterating the former aspect by proposing more sophisticated uncertainty estimation algorithms, and not enough emphasis has been placed on establishing their utility and reproducibility in the very safety-critical settings they were designed for.

The paper is well-written and I found the empirics to be thorough and convincing. I believe this work will help guide the BDL community toward more practicable outcomes so I recommend acceptance.

**Weaknesses:**

One weakness of this type of empirical review is that - although appropriately broad and comprehensive for a conference submission - the empirical study can always be extended to other tasks which may or may not change the conclusions. Another weakness is the lack of novelty from a methodological perspective; although I do appreciate the proposed usage of a signed calibration error to identify under- or overconfidence without having to look at reliability diagrams.

I would suggest including an itemized list of the main takeaways from the empirical study in the conclusion/discussion section of this paper. I think this would help organize the findings of the study and would provide easy-to-follow actionable advice for BDL practitioners going forward.

**Questions:**

-Can you summarise the implications for practitioners who use large-scale models and need to make decisions under distribution shifts?\
-Do you believe there is an inherent trade-off between accuracy, efficiency and calibration? \
-In your opinion, what are the key research directions to further improve the generalization and calibration capabilities of BDL in real-world distribution-shifted data scenarios?\
-How do frequentist methods like conformal prediction factor into this study?

**Limitations:**

See the weaknesses section above.

-Figures 3 and 4 are a bit too small, consider using a horizontal legend for example.

---

> ### Author Rebuttal · Authors · 2023-08-09
>
> We thank you very much for your helpful and positive feedback. We are thrilled that you find that our paper "makes real progress" towards practically applicable BDL and that it is "thorough and convincing". We answer your remaining questions below. Please also see the global response for a list of takeaway messages, results for SNGP, and a list of all citations.
>
> **Itemized list of takeaways and implications for practitioners.** Thank you very much for the suggestion. We added a list of takeaway messages to the conclusion and provide it here on OpenReview as part of the global response.
>
> **Trade-off between accuracy, efficiency, and calibration.** Given our experiments, we do see a "pick only two of the three options" situation in the current state of the art. Given our current understanding of BDL, accurate posterior approximations should achieve accuracy and calibration gains, but are currently very expensive to obtain. However, we think it is conceivable that future algorithms may allow cheap but accurate Bayesian inference.
>
> **Key research directions.** We think that not only from our research it has become clear that multi-modal posterior approximations are central to BDL. While MultiX works well, it is computationally expensive and therefore unlikely to be of practical use in the current world of models with ever-growing parameter counts. Therefore, we think cheap multi-modal posterior approximations are one of the most important future research directions. In a similar spirit, we think post-hoc BDL algorithms such as the Laplace approximation have significant potential to be used in practice to provide uncertainty estimates on top of already trained models. More broadly, finetuning with Bayesian principles as evaluated by us is promising due to its relative computational cheapness, but requires further research, maybe even in the form of BDL algorithms specifically designed for finetuning. We will include some discussion along these lines in the conclusion of the paper revision.
>
> **Frequentist methods.** Frequentist methods are certainly an important and in many cases well working approach to uncertainty quantification. However, we see them as an orthogonal approach to the Bayesian methods that we evaluate, which build on the idea of inferring a posterior over the neural network's parameters. For practical applications of uncertainty quantification, of course all types of methods should be considered, including frequentist/deterministic uncertainty quantification methods. Please also see our global response for an extended discussion of our algorithm selection.
>
> Finally, we appreciate your suggestions regarding Figures 3 and 4 and will include this feedback in our revision.

---

> > ### Comment · Reviewer_Sjhm · 2023-08-16
> > **Response to the Rebuttal**
> >
> > I thank the authors for the detailed response, I am happy with the answers and retain my original score. I would only like to re-emphasize the need to include a list of easily digestible takeaway findings in the discussion/conclusion section of the paper.

---

> > > ### Author Response · Authors · 2023-08-21
> > >
> > > Thank you again very much for your helpful and positive feedback! We will make sure to add the takeaway messages to the conclusion of the revised manuscript.

---

### Official Review · Reviewer_1hhR · 2023-07-19

**Soundness:** 3 good
**Presentation:** 3 good
**Contribution:** 2 fair
**Rating:** 7
**Confidence:** 3

**Summary:**

The paper conducts a large-scale benchmark of Bayesian deep learning (BDL) methods for distribution-shifted data. The focus is on evaluating the quality of the posterior approximation, generalization ability, and calibration using signed versions of calibration metrics to distinguish between under- and overconfidence. The BDL algorithms are evaluated on convolutional and transformer-based architectures, mostly on regression and classification tasks from the WILDS collection. For most tasks, the experiments show improved generalization and calibration when extending single-mode approximations to multiple modes through ensembling. The authors identify a limitation of ensemble-based methods when finetuning transformer-based language models. In this task, accuracy and calibration are not significantly improved over single-mode approximations.

**Strengths:**

Although there are similar experimental surveys of BDL algorithms, the paper benchmarks a wider range of state-of-the-art BDL algorithms applied to modern neural network architectures. Considering the success of ensembling in applications and recent competitions, the choice of focusing on the evaluation of ensembling single-mode BDL algorithms is relevant. The lack of improvement in the generalization and calibration of such methods in transformer-based finetuning tasks is insightful. The introduced signed calibration metrics are useful for making the distinction between over- and underconfident models. The authors have included code with their submission, which is useful for reviewing the details of the experiments and algorithmic implementations. The experiments are described with sufficient clarity and detail.

**Weaknesses:**

1) The paper focuses largely on finetuning tasks, with the exceptions of corrupted CIFAR-10 and the large-scale regression task. The limitation should be described in the “Limitations” section or amended by including further experiments of models that are trained from scratch. Some of the algorithms have specific weaknesses or strengths that become only evident in such settings. For instance, ensemble-based methods may become computationally restrictive. Methods like iVON, SVGD, and BBB that modify the training objective or introduce noise to regularize the training may lead to base models with improved generalization or calibration. When focusing on finetuning of pre-trained models, these effects are neglected to some extent.

2) The experiment on the RXRX1- WILDS datasets in Fig. 3 lacks results for several algorithms. The figure caption and main text do not justify this choice. The exception is the Laplace approximation, where underperformance is noted as a reason.


**Questions:**

1) In the conclusion, the authors state that “[...] we demonstrated that BDL is in many cases competitive with algorithms that are specifically designed for o.o.d. generalization.” To justify that conclusion, additional comparisons to such algorithms, either from existing works or through experimental evaluation, should be added. Except for a hint to the WILDS leaderboard for the large-scale regression experiments, comparisons to specialized methods that are designed for o.o.d. generalization are missing from the main text.

2) One main result of the paper is that ensembling-based BDL methods do not systematically improve upon the single-mode counterparts in finetuning of transformers-based language models. The authors hypothesize that this finding may be due to the nature of the finetuning task. There, the initialization from a pre-trained model may lead to a lack of diversity among the ensemble members. Considering the results of the “Finetuning for CNNs” section, where ensemble methods yield clear improvements, the hypothesis is not fully supported. To clarify the reason for the lack of performance of ensemble-based methods, results on a second architecture in addition to DistillBERT would be insightful. Experimental results to identify the mechanism among task, architecture (BERT-specific?), training procedure, or others, would further strengthen the submission.

---

> ### Author Rebuttal · Authors · 2023-08-09
>
> We thank you a lot for your helpful and positive feedback. We are happy that you find our choice of algorithms to be "relevant", and the failure of ensembles on transformer-based finetuning tasks to be "insightful". We answer your remaining questions below. Please also see the global response for a list of takeaway messages, results for SNGP, and a list of all citations.
>
> **Finetuning tasks.** We added the following paragraph to the limitations:
> *Except for the results on CIFAR-10-(C) and PovertyMap-wilds, all results were obtained by finetuning pre-trained models and are therefore only valid in this setting.*
> Note that even when finetuning, training an ensemble requires training multiple models independently with the associated computational cost, since we finetune all layers of the models. Of course, training is still significantly less computationally expensive than when training all members from scratch.
>
> **Missing results for RxRx1 in Figure 3.** We excluded the VI algorithms for the same reason as Laplace: They are significantly less accurate than the other methods and including them in the figure would make the figure hardly readable.
> The results can be found in the appendix in Figure 14. We added a reference to this figure in the caption of Figure 3 and made the reason for exclusion clearer.
>
> **Comparison with additional baselines.** Thank you very much for the suggestion. Since the WILDS leaderboard [3] contains extensive baselines for all WILDS tasks, including the classification tasks, we will include a small selection of results from WILDS and a more prominent reference to the leaderboard to better support our conclusion.
>
> **Additional results on transformers other than BERT.** Thank you very much for the suggestion. We agree with you that results on an additional architecture would be helpful, however we do find this to be infeasible given the significant computational cost of training ensembles. We decided on the two WILDS tasks due to the availability of baselines for this tasks on the WILDS leaderboard [3]. However, we do have additional results available for models trained with only very small regularization, as we hypothesized that this would allow the ensemble members to better diversify. Please see Figure 3 in the uploaded PDF. On CivilComments, we found that the accuracy of MAP increases when only using a weight decay factor of $10^{-4}$ (though not to the accuracy level of BBB), but the Deep Ensemble still does not perform better than a single model.

---

### Official Review · Reviewer_CBPk · 2023-07-25

**Soundness:** 3 good
**Presentation:** 3 good
**Contribution:** 2 fair
**Rating:** 6
**Confidence:** 3

**Summary:**

This paper compares a variety of Bayesian Deep Learning methods on the WILDS dataset collection comprised of regression and classification tasks with the aim of evaluation generalization performance under distribution shifts. In addition to previously proposed single-mode approximation methods, the authors extend the methodology to sampling ensembles of these kinds of methods and show an improvement on various tasks. Additionally, the authors adapt existing calibration metrics to a signed version that is able to reflect under- or over confidence more directly.

General comment:
- As you state, your paper is in line with the work of (Band et al. 2021) which was submitted to and accepted at the Neurips Benchmark track. The [Neurips website](https://nips.cc/Conferences/2023/CallForDatasetsBenchmarks) states that it is suited among other criteria for "Systematic analyses of existing systems on novel datasets yielding important new insight." which to me perfectly describes this work and in conclusion seems to be a better fit than the main track.

**Strengths:**

- extensive experiments across image classification and regression tasks as well as text based classification
- released code can be run without errors and provides adequate comments about running the code
- the methodology of the paper builds on established datasets and methods and provides a useful benchmark of BDL on more realistic datasets

**Weaknesses:**

- I believe the paper could benefit from a small discussion about why Bayesian Deep Learning in general should be well suited to OOD tasks since Bayesian methods through Bayes rule assume that the test data follows the same distribution as the training data. A relevant discussion seems to be this paper for example (https://arxiv.org/abs/2110.06020)
- I think the training/evaluation setup over the different folds and how those folds are created should be explained more explicitly
- The wide variety of methods you compare is a great strength of the paper, however, I was wondering why you did not include Deep Kernel Learning (DKL) ([Wilson et al. 2015](https://arxiv.org/abs/1511.02222)) or an "improvement" of it like DUE ([Amersfoort et al. 2021](https://arxiv.org/abs/2102.11409)) since their experiments show good generalization performance on a variety of tasks and are also theoretically grounded in Gaussian Processes which many regard as a "gold standard" in Bayesian Machine Learning and UQ
- you state that you provide the first systematic evaluation of BDL for finetuning large pre-trained models, but I would argue that the common practice of using imagenet weights for example is also finetuning? Or are you saying your experiments are the first systematic evaluation with respect to binary text classification?
- the performance of "generalizability" seems to focus on accuracy metrics such as the pearson coefficient or F1 score for classification, however, isn't a more suitable metric the NLL or other proper scoring rules [Gneiting and Raferty 2007](https://www.tandfonline.com/doi/abs/10.1198/016214506000001437) because accuracy metrics ignore the predictive uncertainty? I am aware that these are provided for the regression task in the appendix but should they not be part of the main analysis?

Extra comments:
- On some tables in the Appendix I am not clear about the use of bold face, for example in Table 8 the entire MSE column is bold, or I misread the table?

**Questions:**

- I am slightly confused by how the methodology and code defines a predictive distribution or predictive uncertainty. It appears that you use a Gaussian Mixture Model approach for the Deep Ensembles, but in the appendix you state that "We optimize the log likelihood of the training data and use a fixed standard deviation of 0.1, as this is the value MAP converges to when jointly optimizing the standard deviation and the model's parameters". Is this supposed to be an estimate of the aleatoric uncertainty that is added to the epistemic uncertainty you obtain when computing a variance over the sampled point predictions?
- Why can a pretrained model like resnet18 not be used for the povertyMAP task, but instead you train a model from scratch? At least I was not able to find loading an image-net pretrained weights in the code and there is lots of papers in the earth observation domain that show that image-net outperforms training from scratch even on satellite based optical imagery (as a latest example [Corley et al 2023](https://arxiv.org/pdf/2305.13456.pdf))
- from my understanding a strength of the Laplace approximation is that it yields an uncertainty estimate on top of the MAP estimate and it should theoretically yield the same mean prediction as the MAP model which should be reflected in the same accuracy metric, however, this is not the case in almost all experiments, like Figure 2 and 3. I understand that you are using a sampling approach but perhaps you are not using enough samples? A striking example seems Figure 3 where you state that Laplace underperforms on FMoW and RxRx1 while a Deep Ensemble that is also based on MAP estimates performs well.
- As you state in your conclusion, BBB performs quiet well on a variety of tasks but it appears they are always beat by the much simpler MC-Dropout approach in terms of the TV metric when compared to the gold standard HMC uncertainty estimation (Figure 5). I was wondering whether you could comment on this discrepancy and whether you have a possible explanation for this?

**Limitations:**

The limitations are accurately addressed.

---

> ### Author Rebuttal · Authors · 2023-08-09
>
> Thank you very much for your constructive and detailed feedback. We hope the following points help to resolve all your questions regarding our work. If you have any further questions, please let us know so that we can clarify things further. Please also see the global response for a list of takeaway messages, results for SNGP, and a list of all citations.
>
> **NeurIPS Benchmark track.** We agree with you that the Benchmark track would have also been a good fit for our paper. Before submitting the paper, we reviewed both the main track and the Benchmark track and decided for the main track, since the NeurIPS Call for Papers for the main track explicitly lists "Evaluation (e.g., methodology, meta studies, replicability and validity)" as a submission topic and the Benchmark track's Call for Papers states "It is also still possible to submit datasets and benchmarks to the main conference".
>
> **Discussion about the usefulness of BDL.** Thank you very much for the suggestion. We added the proposed discussion to the introduction, mentioning both the theoretical results of [12], the intuition about the Bayesian Model Average given by [4], and the good OOD calibration results that BDL can achieve in practice as reported by practical evaluations [1, 4, 6], including our own.
>
> **Training/Evaluation Setup.** We added an explicit description of the folds and training/test setup to Appendix G.3. For all WILDS tasks, we used the same splits as [3] to ensure that our results are directly comparable with their results. We used in-distribution validation splits and, where available, OOD validation splits for hyperparameter optimization, in line with the protocol of [3].
>
> **Deep Kernel Learning.** Thank you for the suggestion! We added SNGP, a scalable variant of DKL, as an additional baseline. Please see our global response for further details.
>
> **Initialization from ImageNet weights.** We agree with you that initializing from ImageNet weights can also be considered finetuning. However, to the best of our knowledge, our evaluation is the first to systematically evaluate finetuning across a wide range of BDL algorithms and tasks. For example, neither of the BDL evaluations of [8] and [5] consider finetuning, and [2] do so only in a very limited fashion. We clarify this point in the revised version of our manuscript.
>
> **NLL Metric.** We decided to specifically separate predictive accuracy and uncertainty, as we find the NLL to be hard to interpret, given that it mixes both the accuracy and calibration into a single metric. Especially when considering practical applications of BDL, we believe that it is relevant to be able to assess accuracy and uncertainty independently, which is why we decided to provide these metrics in the main text and move the NLL to the appendix. In our revised paper, we more prominently reference the NLL and also include it for all tasks where it is not yet present in Appendix G. We also provide the extended tables for FMoW and CivilComments in the uploaded PDF file (Table 1).
>
> **Bold face in tables.** The bold face is intended to highlight the best performing methods within two times standard error. However, there are a few tables in the appendix where we incorrectly highlighted all numbers. Thanks for catching this! We will fix it in the revised version.
>
> **Predictive Distribution & Ensembles.** As you correctly state, we treat ensembles as mixture models where each mixture component receives equal weight. The standard deviation in the NLL of PovertyMap does indeed represent the aleatoric uncertainty. We clarified this in Appendix G.3.2.
>
> **Pretraining on PovertyMap.** We agree with you that a pretrained model would most likely have improved the model's accuracy. However, our goal was to stay consistent with [3] whenever possible, so that our results can be fairly compared to their non-Bayesian baselines. In particular, [3] do not use a pretrained model on PovertyMap (see Appendix E.8.1 of [3]).
>
> **Performance of Laplace.** Following your remarks, we add an extended discussion of how we made sure that our implementation works correctly in our revised manuscript and also in the uploaded PDF file. To check for sampling issues, we evaluated Laplace with 1000 posterior samples on FMoW and 100 on RxRx1 and found an increase in accuracy, with Laplace still performing worse than MAP (see Figure 2 in the uploaded PDF file). We will rerun the Laplace evaluations for all tasks and include the results in the revision of the paper, with an explicit mention of the different number of posterior samples. Further, we will add a discussion of the sampling issues in the appendix. To further check our implementation, we evaluated the Laplace model at its parameter posterior mean on FMoW and recovered the accuracy of MAP.
>
> **BBB vs. MCD.** The agreement with HMC reported in Figure 9 and Table 7 in our appendix may give an insight into this: BBB's and HMC's predictions more frequently differ from each other than MCD's and HMC's, which leads to a high TV in case of differing predictions. Further, our evaluation shows that the results on CIFAR, which is the only task where HMC samples are available to evaluate the TV, cannot be transferred to the text classification tasks where BBB performs particularly well compared to MCD.

---

> > ### Comment · Reviewer_CBPk · 2023-08-14
> >
> > Thank you for your elaborate response to the raised questions and the effort put forward around additional experiments and improvements to the manuscript and therefore raise my score.

---

> > > ### Author Response · Authors · 2023-08-14
> > >
> > > Thank you very much for your response and for raising the score. We are happy that we were able to address your concerns.

---

### Author Rebuttal · Authors · 2023-08-09

We would like to thank all reviewers for their time and efforts in providing detailed and insightful feedback, which we will incorporate in our revision.

**We are pleased that the reviewers are convinced that our paper "provides a useful benchmark of BDL on more realistic datasets" (Reviewer CBPk) and is "appropriately broad and comprehensive for a conference submission" (Reviewer Sjhm). The reviewers think our results are "insightful" (Reviewer 1hhR) and "convincing" (Reviewer Sjhm), and that the paper "makes real progress" (Reviewer Sjhm) towards real-world applicability of BDL.**

Reviewer eTS5 was hesitant to recommend acceptance due to similarities to other BDL evaluation papers and a focus on the evaluation itself, rather than a deep investigation of singular results. Therefore, we would like to emphasize that our work contains a number of novel results which we judge to be highly relevant for practitioners as summarized below.

## Key Takeaways
As requested by reviewers Sjhm, hcj9, and eTS5, we would like to summarize our key takeaways, which we expect to have direct implications for practitioners and method developers and have been unknown or not thoroughly shown until now. Since we evaluated on diverse, realistic distribution-shifted data, the results are likely directly applicable to many real-world applications of BDL.

- Finetuning only the last layers of pre-trained models with BDL algorithms gives a significant boost of generalization accuracy and calibration on realistic distribution-shifted data, while incurring a comparatively small runtime overhead. These models are in many cases competitive to or even outperform methods that are specially designed for OOD generalization such as IRM [12] and Fish [13].
- For CNNs, ensembles are more accurate and better calibrated on OOD data than single-mode posterior approximations by a wide margin, even when initializing all ensemble members from the same pre-trained checkpoint with only the last layers differently initialized, i.e. when not using the standard protocol of randomly initializing all ensemble members. Ensembling probabilistic single-mode posterior approximations such as SWAG or MCD yields only a small additional increase in accuracy and calibration.
- When finetuning large transformers, ensembles yield no benefit. Compared to all other evaluated BDL algorithms, classical mean-field variational inference achieves significant accuracy gains under distribution shift. We want to emphasize the importance of this point: Our results show that ensembles, which are typically considered to be the SOTA in BDL, do not work well for finetuning large transformers, a setting that is highly relevant given the recent success of large language models.

In the paper revision, we added these points prominently as part of the conclusion of the paper.

## Algorithm Selection
**We follow the requests of reviewers eTS5, CBPk, and hcj9 and add SNGP [10], a scalable variant of DKL, as an additional baseline for the WILDS tasks. We provide preliminary results in the additionally uploaded PDF file (Figure 1).** Please note that these results are not directly comparable to our other results and the WILDS baselines due to necessary modifications of the architecture, in particular the addition of a GP. Due to the limited timeframe, we could so far only evaluate with the default hyperparameters from [19]. We are working on completing the results for all WILDS datasets.

Furthermore, as multiple reviewers requested additional algorithms, we want to clarify our selection of algorithms. We selected the algorithms based on the following criteria, that ensure scalability of the algorithms and allow a fair comparison with the non-Bayesian baselines of WILDS:

- Algorithms must scale to large parameter counts and datasets. This excludes MCMC-based methods.
- It must not be required to modify the underlying neural network. This ensures that the comparison with the WILDS baselines is fair, and excludes GP-based methods such as DKL, SNGP, and DUE.
- Algorithms must approximate the parameter posterior of the underlying neural network. This allows us to compare the algorithms with the HMC samples from [6], and excludes deterministic UQ methods such as DUQ.
- Algorithms must be applicable to both classification and regression problems. This excludes e.g. fBNN.

We belief that these criteria make for a fair comparison that is relevant to practitioners. In particular, reviewers hcj9, Sjhm, and 1hhR agree with us on the relevance of the selected algorithms.

[1] Schneider, Steffen et al. "Improving robustness against common corruptions by covariate shift adaptation" NeurIPS 2020

[2] Nado, Zachary, et al. "Uncertainty Baselines: Benchmarks for uncertainty \& robustness in deep learning." arXiv 2021

[3] Koh, Pang Wei, et al. "WILDS: A Benchmark of in-the-Wild Distribution Shifts" ICML 202

[4] Wilson, Andrew G., et al. "Bayesian Deep Learning and a Probabilistic Perspective of Generalization" NeurIPS 2020

[5] Band, Neil, et al. "Benchmarking Bayesian Deep Learning on Diabetic Retinopathy Detection Tasks" NeurIPS 2021 Datasets and Benchmarks Track

[6] Izmailov, Pavel, et al. "What Are Bayesian Neural Network Posteriors Really Like?", ICML 2021

[7] Sharma, Mrinank, et al. "Do Bayesian Neural Networks Need To Be Fully Stochastic?" AISTATS 2023

[8] Ovadia, Yaniv, et al. "Can you trust your model's uncertainty? Evaluating predictive uncertainty under dataset shift" NeurIPS 2019

[9] Daxberger, Erik, et al. "Laplace redux - effortless bayesian deep learning" NeurIPS 2021

[10] Liu, Jeremiah, et al. "Simple and principled uncertainty estimation with deterministic deep learning via distance awareness" NeurIPS 2020

[11] D'Angelo et al. "On out-of-distribution detection with Bayesian neural networks" arXiv 2022

[12] Arjovsky, Martin, et al. "Invariant risk minimization" arXiv 2019

[13] Yuge, Shi et al. "Gradient Matching for Domain Generalization" ICLR 2022

---

### Author Response · Authors · 2023-08-21

We would like to thank the reviewers again for their effort in reviewing our paper, which allowed us to further improve our paper. In particular, we included SNGP as an additional baseline, and added concise takeaway messages to the conclusion of the paper. These changes led to reviewers CBPk and eTS5 increasing their scores, with now four out of five reviewers recommending acceptance.

---

### Decision · Program_Chairs · 2023-09-21

**Decision:**

Accept (poster)

**Comment:**

This paper performs a large-scale experimental study of Bayesian NNs under distribution shift, focusing on the WILDS benchmarks.  Of particular note are the experiments involving transformers and multi-modal (i.e. ensembled) versions of popular techniques (e.g. Laplace).  The work reports findings such as the benefits of last-layer fine-tuning, CNN's amenability to ensembling, and the lack of benefit of ensembles of transformers.  The reviewers are generally in favor of acceptance (1 x strong accept, 1 x accept, 2 x weak accept, 1 x borderline reject).  Reviewer eT5S's, the only to favor rejection, primary criticisms are (i) lack of contribution due to no novel methodology, (ii) other papers have previously evaluated on WILDS, and (iii) lack of clear takeaways / recommendations.  While these criticisms do have some merits, I side with the other reviewers that the scale of the experimental evaluation, including the aforementioned findings, does make for a sufficiently interesting contribution.  Moreover, the work goes well beyond previous work that has evaluated on WILDS and performed other largescale benchmarking, e.g. with the analysis of transformers and of multi-modal approximations.

The authors should address the above concerns in the camera-ready version, as promised.  Moreover, many of the plots can be improved in the size of their text and general format.  They are generally small and hard to read without zooming into the pdf.